# Adversarial Graph Augmentation to Improve Graph Contrastive Learning

**Susheel Suresh**
Purdue University
suresh43@purdue.edu

**Pan Li**[*]
Purdue University
panli@purdue.edu

**Cong Hao**
Georgia Tech
callie.hao@gatech.edu

**Jennifer Neville**
Purdue University and Microsoft Research
jenneville@microsoft.com

## Abstract

Self-supervised learning of graph neural networks (GNN) is in great need because of the widespread label scarcity issue in real-world graph/network data. Graph contrastive learning (GCL), by training GNNs to maximize the correspondence between the representations of the same graph in its different augmented forms, may yield robust and transferable GNNs even without using labels. However, GNNs trained by traditional GCL often risk capturing redundant graph features and thus may be brittle and provide sub-par performance in downstream tasks. Here, we propose a novel principle, termed adversarial-GCL (*AD-GCL*), which enables GNNs to avoid capturing redundant information during the training by optimizing adversarial graph augmentation strategies used in GCL. We pair AD-GCL with theoretical explanations and design a practical instantiation based on trainable edge-dropping graph augmentation. We experimentally validate AD-GCL[2] by comparing with the state-of-the-art GCL methods and achieve performance gains of up-to 14% in unsupervised, 6% in transfer, and 3% in semi-supervised learning settings overall with 18 different benchmark datasets for the tasks of molecule property regression and classification, and social network classification.

## 1 Introduction

Graph representation learning (GRL) aims to encode graph-structured data into low-dimensional vector representations, which has recently shown great potential in many applications in biochemistry, physics and social science [1–3]. Graph neural networks (GNNs), inheriting the power of neural networks [4, 5], have become the almost *de facto* encoders for GRL [6–9]. GNNs have been mostly studied in cases with supervised end-to-end training [10–16], where a large number of task-specific labels are needed. However, in many applications, annotating labels of graph data takes a lot of time and resources [17, 18], e.g., identifying pharmacological effect of drug molecule graphs requires living animal experiments [19]. Therefore, recent research efforts are directed towards studying self-supervised learning for GNNs, where only limited or even no labels are needed [18, 20–31].

Designing proper self-supervised-learning principles for GNNs is crucial, as they drive what information of graph-structured data will be captured by GNNs and may heavily impact their performance in downstream tasks. Many previous works adopt the edge-reconstruction principle to match traditional network-embedding requirement [32–35], where the edges of the input graph are expected to be reconstructed based on the output of GNNs [20, 21, 36]. Experiments showed that these GNN models learn to over-emphasize node proximity [23] and may lose subtle but crucial structural information, thus failing in many tasks including node-role classification [16, 35, 37, 38] and graph classification [17].

---

[*]Pan Li and Jennifer Neville co-correspond this work.
[2]https://github.com/susheels/adgcl

35th Conference on Neural Information Processing Systems (NeurIPS 2021).

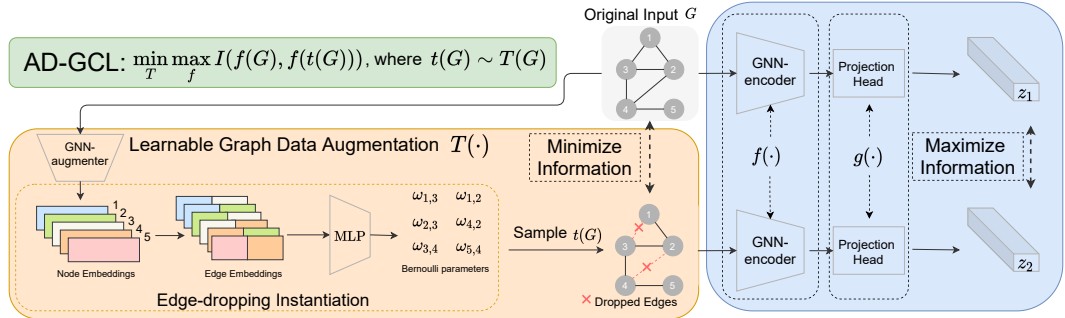

Figure 1: The AD-GCL principle and its instantiation based on learnable edge-dropping augmentation. AD-GCL contains two components for graph data encoding and graph data augmentation. The GNN encoder $f(\cdot)$ maximizes the mutual information between the original graph $G$ and the augmented graph $t(G)$ while the GNN augmenter optimizes the augmentation $T(\cdot)$ to remove the information from the original graph. The instantiation of AD-GCL proposed in this work uses edge dropping: An edge $e$ of $G$ is randomly dropped according to Bernoulli($\omega_e$), where $\omega_e$ is parameterized by the GNN augmenter.

To avoid the above issue, graph contrastive learning (GCL) has attracted more attention recently [18, 22, 23, 25–31]. GCL leverages the mutual information maximization principle (InfoMax) [39] that aims to maximize the correspondence between the representations of a graph (or a node) in its different augmented forms [18, 24, 25, 28–31]. Perfect correspondence indicates that a representation precisely identifies its corresponding graph (or node) and thus the encoding procedure does not decrease the mutual information between them.

However, researchers have found that the InfoMax principle may be risky because it may push encoders to capture redundant information that is irrelevant to the downstream tasks: Redundant information suffices to identify each graph to achieve InfoMax, but encoding it yields brittle representations and may severely deteriorate the performance of the encoder in the downstream tasks [40]. This observation reminds us of another principle, termed information bottleneck (IB) [41–46]. As opposed to InfoMax, IB asks the encoder to capture the *minimal sufficient* information for the downstream tasks. Specifically, IB minimizes the information from the original data while maximizing the information that is relevant to the downstream tasks. As the redundant information gets removed, the encoder learnt by IB tends to be more robust and transferable. Recently, IB has been applied to GNNs [47, 48]. But IB needs the knowledge of the downstream tasks that may not be available.

Hence, a natural question emerges: *When the knowledge of downstream tasks are unavailable, how to train GNNs that may remove redundant information?* Previous works highlight some solutions by designing data augmentation strategies for GCL but those strategies are typically task-related and sub-optimal. They either leverage domain knowledge [25, 28, 30], *e.g.*, node centralities in network science or molecule motifs in bio-chemistry, or depend on extensive evaluation on the downstream tasks, where the best strategy is selected based on validation performance [24, 30].

In this paper, we approach this question by proposing a novel principle that pairs GCL with adversarial training, termed *AD-GCL*, as shown in Fig.1. We particularly focus on training self-supervised GNNs for graph-level tasks, though the idea may be generalized for node-level tasks. AD-GCL consists of two components: The first component contains a GNN encoder, which adopts InfoMax to maximize the correspondence/mutual information between the representations of the original graph and its augmented graphs. The second component contains a GNN-based augmenter, which aims to optimize the augmentation strategy to decrease redundant information from the original graph as much as possible. AD-GCL essentially allows the encoder capturing the minimal sufficient information to distinguish graphs in the dataset. We further provide theoretical explanations of AD-GCL. We show that with certain regularization on the search space of the augmenter, AD-GCL can yield a lower bound guarantee of the information related to the downstream tasks, while simultaneously holding an upper bound guarantee of the redundant information from the original graphs, which matches the aim of the IB principle. We further give an instantiation of AD-GCL: The GNN augmenter adopts a task-agnostic augmentation strategy and will learn an input-graph-dependent non-uniform-edge-drop probability to perform graph augmentation.

Finally, we extensively evaluate AD-GCL on 18 different benchmark datasets for molecule property classification and regression, and social network classification tasks in different setting viz. unsuper-

vised learning (Sec. 5.1), transfer learning (Sec. 5.3) and semi-supervised learning (Sec. 5.4) learning. AD-GCL achieves significant performance gains in relative improvement and high mean ranks over the datasets compared to state-of-the-art baselines. We also study the theoretical aspects of AD-GCL with apt experiments and analyze the results to offer fresh perspectives (Sec. 5.2): Interestingly, we observe that AD-GCL outperforms traditional GCL based on non-optimizable augmentation across almost the entire range of perturbation levels.

## 2 Notations and Preliminaries

We first introduce some preliminary concepts and notations for further exposition. In this work, we consider attributed graphs $G = (V, E)$ where $V$ is a node set and $E$ is an edge set. $G$ may have node attributes $\{X_v \in \mathbb{R}^F \mid v \in V\}$ and edge attributes $\{X_e \in \mathbb{R}^F \mid e \in E\}$ of dimension $F$. We denote the set of the neighbors of a node $v$ as $\mathcal{N}_v$.

**Learning Graph Representations.** Given a set of graphs $G_i$, $i = 1, 2, ..., n$, in some universe $\mathcal{G}$, the aim is to learn an encoder $f : \mathcal{G} \to \mathbb{R}^d$, where $f(G_i)$ can be further used in some downstream task. We also assume that $G_i$'s are all IID sampled from an unknown distribution $\mathbb{P}_\mathcal{G}$ defined over $\mathcal{G}$. In a downstream task, each $G_i$ is associated with a label $y_i \in \mathcal{Y}$. Another model $q : \mathbb{R}^d \to \mathcal{Y}$ will be learnt to predict $Y_i$ based on $q(f(G_i))$. We assume $(G_i, Y_i)$'s are IID sampled from a distribution $\mathbb{P}_{\mathcal{G} \times \mathcal{Y}} = \mathbb{P}_{\mathcal{Y}|\mathcal{G}} \mathbb{P}_\mathcal{G}$, where $\mathbb{P}_{\mathcal{Y}|\mathcal{G}}$ is the conditional distribution of the graph label in the downstream task given the graph.

**Graph Neural Networks (GNNs).** In this work, we focus on using GNNs, message passing GNNs in particular [49], as the encoder $f$. For a graph $G = (V, E)$, every node $v \in V$ will be paired with a node representation $h_v$ initialized as $h_v^{(0)} = X_v$. These representations will be updated by a GNN. During the $k^{\text{th}}$ iteration, each $h_v^{(k-1)}$ is updated using $v'$s neighbourhood information expressed as,

$$h_v^{(k)} = \text{UPDATE}^{(k)}\left( h_v^{(k-1)}, \ \text{AGGREGATE}^{(k)}\left( \{(h_u^{(k-1)}, X_{uv}) \mid u \in \mathcal{N}_v\} \right) \right) \tag{1}$$

where AGGREGATE$(\cdot)$ is a trainable function that maps the set of node representations and edge attributes $X_{uv}$ to an aggregated vector, UPDATE$(\cdot)$ is another trainable function that maps both $v$'s current representation and the aggregated vector to $v$'s updated representation. After $K$ iterations of Eq. 1, the graph representation is obtained by pooling the final set of node representations as,

$$f(G) :\triangleq h_G = \text{POOL}\left( \{h_v^{(K)} \mid v \in V\} \right) \tag{2}$$

For design choices regarding aggregation, update and pooling functions we refer the reader to [3, 7, 8].

**The Mutual Information Maximization Principle.** GCL is built upon the InfoMax principle [39], which prescribes to learn an encoder $f$ that maximizes the mutual information or the correspondence between the graph and its representation. The rationale behind GCL is that a graph representation $f(G)$ should capture the features of the graph $G$ so that representation can distinguish this graph from other graphs. Specifically, the objective of GCL follows

$$\text{InfoMax:} \quad \max_f I(G; f(G)), \quad \text{where } G \sim \mathbb{P}_\mathcal{G}. \tag{3}$$

where $I(X_1; X_2)$ denotes the mutual information between two random variables $X_1$ and $X_2$ [50].

Note that the encoder $f(\cdot)$ given by GNNs is not injective in the graph space $\mathcal{G}$ due to its limited expressive power [14, 15]. Specifically, for the graphs that cannot be distinguished by 1-WL test [51], GNNs will associate them with the same representations. We leave more discussion on 1-WL test in Appendix C. In contrast to using CNNs as encoders, one can never expect GNNs to identify all the graphs in $\mathcal{G}$ based their representations, which introduces a unique challenge for GCL.

## 3 Adversarial Graph Contrastive Learning

In this section, we introduce our adversarial graph contrastive learning (AD-GCL) framework and one of its instantiations based on edge perturbation.

### 3.1 Theoretical Motivation and Formulation of AD-GCL

The InfoMax principle in Eq. 3 could be problematic in practice for general representation learning. Tschannen et al. have shown that for image classification, representations capturing the information

that is entirely irrelevant to the image labels are also able to maximize the mutual information but such representations are definitely not useful for image classification [40]. A similar issue can also be observed in graph representation learning, as illustrated by Fig.2: We consider a binary graph classification problem with graphs in the dataset ogbg-molbace [52]. Two GNN encoders with exactly the same architecture are trained to keep mutual information maximization between graph representations and the input graphs, but one of the GNN encoders in the same time is further supervised by random graph labels. Although the GNN encoder supervised by random labels still keeps one-to-one correspondance between every input graph and its representation (i.e., mutual information maximization), we may observe significant performance degeneration of this GNN encoder when evaluating it over the downstream ground-truth labels. More detailed experiment setup is left in Appendix G.1.

This observation inspires us to rethink what a good graph representation is. Recently, the information bottleneck has applied to learn graph representations [47, 48]. Specifically, the objective of graph information bottleneck (GIB) follows

$$\text{GIB:} \quad \max_f I(f(G); Y) - \beta I(G; f(G)), \quad (4)$$

where $(G, Y) \sim \mathbb{P}_{\mathcal{G} \times \mathcal{Y}}, \beta$ is a positive constant. Comparing Eq. 3 and Eq. 4, we may observe the different requirements between InfoMax and GIB: InfoMax asks for maximizing the information from the original graph, while GIB asks for minimizing such information but simultaneously maximizing the information that is relevant to the downstream tasks. As GIB asks to remove redundant information, GIB naturally avoids the issue encountered in Fig.2. Removing extra information also makes GNNs trained w.r.t. GIB robust to adverserial attack and strongly transferrable [47, 48].

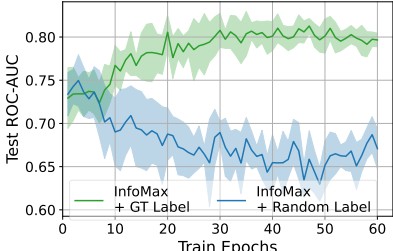

Figure 2: Two GNNs keep the mutual information maximized between graphs and their representations. Simultaneously, they get supervised by ground-truth labels (green) and random labels (blue) respectively. The curves show their testing performance on predicting ground-truth labels.

Unfortunately, GIB requires the knowledge of the class labels $Y$ from the downstream task and thus does not apply to self-supervised training of GNNs where there are few or no labels. Then, the question is how to learn robust and transferable GNNs in a self-supervised way.

To address this, we will develop a GCL approach that uses adversarial learning to avoid capturing redundant information during the representation learning. In general, GCL methods use graph data augmentation (GDA) processes to perturb the original observed graphs and decrease the amount of information they encode. Then, the methods apply InfoMax over perturbed graph pairs (using different GDAs) to train an encoder $f$ to capture the remaining information.

**Definition 1** (Graph Data Augmentation (GDA)). *For a graph $G \in \mathcal{G}$, $T(G)$ denotes a graph data augmentation of $G$, which is a distribution defined over $\mathcal{G}$ conditioned on $G$. We use $t(G) \in \mathcal{G}$ to denote a sample of $T(G)$.*

Specifically, given two ways of GDA $T_1$ and $T_2$, the objective of GCL becomes

$$\text{GDA-GCL:} \quad \max_f I(f(t_1(G)); f(t_2(G))), \text{ where } G \sim \mathbb{P}_{\mathcal{G}}, t_i(G) \sim T_i(G), i \in \{1, 2\}. \quad (5)$$

In practice, GDA processes are often pre-designed based on either domain knowledge or extensive evaluation, and improper choice of GDA may severely impact the downstream performance [17, 24]. We will review a few GDAs adopted in existing works in Sec.4.

In contrast to previous predefined GDAs, our idea, inspired by GIB, is to *learn* the GDA process (over a parameterized family), so that the encoder $f$ can capture the **minimal information** that is sufficient to identify each graph.

**AD-GCL:** We optimize the following objective, over a GDA family $\mathcal{T}$ (defined below).

$$\text{AD-GCL:} \quad \min_{T \in \mathcal{T}} \max_f I(f(G); f(t(G))), \quad \text{where } G \sim \mathbb{P}_{\mathcal{G}}, t(G) \sim T(G), \quad (6)$$

**Definition 2** (Graph Data Augmentation Family). *Let $\mathcal{T}$ denote a family of different GDAs $T_\Phi(\cdot)$, where $\Phi$ is the parameter in some universe. A $T_\Phi(\cdot) \in \mathcal{T}$ is a specific GDA with parameter $\Phi$.*

The min-max principle in AD-GCL aims to train the encoder such that even with a very aggressive GDA (i.e., where $t(G)$ is very different from $G$), the mutual information / the correspondence

between the perturbed graph and the original graph can be maximized. Compared with the two GDAs adopted in GDA-GCL (Eq.5), AD-GCL views the original graph $G$ as the anchor while pushing its perturbation $T(G)$ as far from the anchor as it can. The automatic search over $T \in \mathcal{T}$ saves a great deal of effort evaluating different combinations of GDA as adopted in [24].

**Relating AD-GCL to the downstream task.** Next, we will theoretically characterize the property of the encoder trained via AD-GCL. The analysis here not only further illustrates the rationale of AD-GCL but helps design practical $\mathcal{T}$ when some knowledge of $Y$ is accessible. But note that our analysis does not make any assumption on the availability of $Y$.

Note that GNNs learning graph representations is very different from CNNs learning image representations because GNNs are never injective mappings between the graph universe $\mathcal{G}$ and the representation space $\mathbb{R}^d$, because the expressive power of GNNs is limited by the 1-WL test [14, 15, 51]. So, we need to define a quotient space of $\mathcal{G}$ based on the equivalence given by the 1-WL test.

**Definition 3** (Graph Quotient Space). *Define the equivalence $\cong$ between two graphs $G_1 \cong G_2$ if $G_1$, $G_2$ cannot be distinguished by the 1-WL test. Define the quotient space $\mathcal{G}' = \mathcal{G}/\cong$.*

So every element in the quotient space, i.e., $G' \in \mathcal{G}'$, is a representative graph from a family of graphs that cannot be distinguished by the 1-WL test. Note that our definition also allows attributed graphs.

**Definition 4** (Probability Measures in $\mathcal{G}'$). *Define $\mathbb{P}_{\mathcal{G}'}$ over the space $\mathcal{G}'$ such that $\mathbb{P}_{\mathcal{G}'}(G') = \mathbb{P}_{\mathcal{G}}(G \cong G')$ for any $G' \in \mathcal{G}'$. Further define $\mathbb{P}_{\mathcal{G}' \times \mathcal{Y}}(G', Y') = \mathbb{P}_{\mathcal{G} \times \mathcal{Y}}(G \cong G', Y = Y')$. Given a GDA $T(\cdot)$ defined over $\mathcal{G}$, define a distribution on $\mathcal{G}'$, $T'(G') = \mathbb{E}_{G \sim \mathbb{P}_{\mathcal{G}}}[T(G)|G \cong G']$ for $G' \in \mathcal{G}'$.*

Now, we provide our theoretical results and give their implication. The proof is in the Appendix B.

**Theorem 1.** *Suppose the encoder $f$ is implemented by a GNN as powerful as the 1-WL test. Suppose $\mathcal{G}$ is a countable space and thus $\mathcal{G}'$ is a countable space. Then, the optimal solution $(f^*, T^*)$ to AD-GCL satisfies, letting $T'^*(G') = \mathbb{E}_{G \sim \mathbb{P}_{\mathcal{G}}}[T^*(G)|G \cong G']$,*

1. *$I(f^*(t^*(G)); G \,|\, Y) \leq \min_{T \in \mathcal{T}} I(t'(G'); G') - I(t'^*(G'); Y)$, where $t'(G') \sim T'(G')$, $t'^*(G') \sim T'^*(G')$, $(G, Y) \sim \mathbb{P}_{\mathcal{G} \times \mathcal{Y}}$ and $(G', Y) \sim \mathbb{P}_{\mathcal{G}' \times \mathcal{Y}}$.*

2. *$I(f^*(G); Y) \geq I(f^*(t'^*(G')); Y) = I(t'^*(G'); Y)$, where $t'^*(G') \sim T'^*(G')$, $(G, Y) \sim \mathbb{P}_{\mathcal{G} \times \mathcal{Y}}$ and $(G', Y) \sim \mathbb{P}_{\mathcal{G}' \times \mathcal{Y}}$.*

The statement 1 in Theorem 1 guarantees a upper bound of the information that is captured by the representations but irrelevant to the downstream task, which matches our aim. This bound has a form very relevant to the GIB principle (Eq.4 when $\beta = 1$), since $\min_{T \in \mathcal{T}} I(t'(G'); G') - I(t'^*(G'); Y) \geq \min_f [I(f(G); G) - I(f(G); Y)]$, where $f$ is a GNN encoder as powerful as the 1-WL test. But note that this inequality also implies that the encoder given by AD-GCL may be worse than the optimal encoder given by GIB ($\beta = 1$). This makes sense as GIB has the access to the downstream task $Y$.

The statement 2 in Theorem 1 guarantees a lower bound of the mutual information between the learnt representations and the labels of the downstream task. As long as the GDA family $\mathcal{T}$ has a good control, $I(t'^*(G'); Y) \geq \min_{T \in \mathcal{T}} I(t'(G'); Y)$ and $I(f^*(G); Y)$ thus cannot be too small. This implies that it is better to regularize when learning over $\mathcal{T}$. In our instantiation, based on edge-dropping augmentation (Sec. 3.2), we regularize the ratio of dropped edges per graph.

### 3.2 Instantiation of AD-GCL via Learnable Edge Perturbation

We now introduce a practical instantiation of the AD-GCL principle (Eq. 6) based on learnable edge-dropping augmentations as illustrated in Fig. 1. (See Appendix D for a summary of AD-GCL in its algorithmic form.) The objective of AD-GCL has two folds: (1) Optimize the encoder $f$ to maximize the mutual information between the representations of the original graph $G$ and its augmented graph $t(G)$; (2) Optimize the GDA $T(G)$ where $t(G)$ is sampled to minimize such a mutual information. We always set the encoder as a GNN $f_\Theta$ with learnable parameters $\Theta$ and next we focus on the GDA, $T_\Phi(G)$ that has learnable parameters $\Phi$.

**Learnable Edge Dropping GDA model $T_\Phi(\cdot)$.** Edge dropping is the operation of deleting some edges in a graph. As a proof of concept, we adopt edge dropping to formulate the GDA family $\mathcal{T}$. Other types of GDAs such as node dropping, edge adding and feature masking can also be paired with our AD-GCL principle. Interestingly, in our experiments, edge-dropping augmentation optimized by AD-GCL has already achieved much better performance than any pre-defined random

GDAs even carefully selected via extensive evaluation [24] (See Sec.5). Another reason that supports edge dropping is due to our Theorem 1 statement 2, which shows that good GDAs should keep some information related to the downstream tasks. Many GRL downstream tasks such as molecule classification only depends on the structural fingerprints that can be represented as subgraphs of the original graph [53]. Dropping a few edges may not change those subgraph structures and thus keeps the information sufficient to the downstream classification. But note that this reasoning does not mean that we leverage domain knowledge to design GDA, as the family $\mathcal{T}$ is still broad and the specific GDA still needs to be optimized. Moreover, experiments show that our instantiation also works extremely well on social network classification and molecule property regression, where the evidence of subgraph fingerprints may not exist any more.

**Parameterizing $T_\Phi(\cdot)$.** For each $G = (V, E)$, we set $T_\Phi(G)$, $T \in \mathcal{T}$ as a random graph model [54, 55] conditioning on $G$. Each sample $t(G) \sim T_\Phi(G)$ is a graph that shares the same node set with $G$ while the edge set of $t(G)$ is only a subset of $E$. Each edge $e \in E$ will be associated with a random variable $p_e \sim$ Bernoulli$(\omega_e)$, where $e$ is in $t(G)$ if $p_e = 1$ and is dropped otherwise.

We parameterize the Bernoulli weights $\omega_e$ by leveraging another GNN, *i.e.,* the *augmenter*, to run on $G$ according to Eq.1 of $K$ layers, get the final-layer node representations $\{h_v^{(K)}|v \in V\}$ and set

$$\omega_e = \text{MLP}([h_u^{(K)}; h_z^{(K)}]), \quad \text{where } e = (u, z) \text{ and } \{h_v^{(K)} \mid v \in V\} = \text{GNN-augmenter}(G) \quad (7)$$

To train $T(G)$ in an end-to-end fashion, we relax the discrete $p_e$ to be a continuous variable in $[0, 1]$ and utilize the Gumbel-Max reparametrization trick [56, 57]. Specifically, $p_e = \text{Sigmoid}((\log \delta - \log(1 - \delta) + \omega_e)/\tau)$, where $\delta \sim$ Uniform$(0, 1)$. As temperature hyper-parameter $\tau \to 0$, $p_e$ gets closer to being binary. Moreover, the gradients $\frac{\partial p_e}{\partial \omega_e}$ are smooth and well defined. This style of edge dropping based on a random graph model has also been used for parameterized explanations of GNNs [58].

**Regularizing $T_\Phi(\cdot)$.** As shown in Theorem 1, a reasonable GDA should keep a certain amount of information related to the downstream tasks (statement 2). Hence, we expect the GDAs in the edge dropping family $\mathcal{T}$ not to perform very aggressive perturbation. Therefore, we regularize the ratio of edges being dropped per graph by enforcing the following constraint: For a graph $G$ and its augmented graph $t(G)$, we add $\sum_{e \in E} \omega_e/|E|$ to the objective, where $\omega_e$ is defined in Eq.7 indicates the probability that $e$ gets dropped.

Putting everything together, the final objective is as follows.

$$\min_\Phi \max_\Theta I(f_\Theta(G); f_\Theta(t(G))) + \lambda_{\text{reg}}\mathbb{E}_G\Big[\sum_{e \in E} \omega_e/|E|\Big], \text{ where } G \sim \mathbb{P}_\mathcal{G}, t(G) \sim T_\Phi(G). \quad (8)$$

Note $\Phi$ corresponds to the learnable parameters of the augmenter GNN and MLP used to derive the $\omega_e$'s and $\Theta$ corresponds to the learnable parameters of the GNN $f$.

**Estimating the objective in Eq.8.** In our implementation, the second (regularization) term is easy to estimate empirically. For the first (mutual information) term, we adopt InfoNCE as the estimator [59–61], which is known to be a lower bound of the mutual information and is frequently used for contrastive learning [40, 59, 62]. Specfically, during the training, given a minibatch of $m$ graphs $\{G_i\}_{i=1}^m$, let $z_{i,1} = g(f_\Theta(G_i))$ and $z_{i,2} = g(f_\Theta(t(G_i)))$ where $g(\cdot)$ is the projection head implemented by a 2-layer MLP as suggested in [62]. With $sim(\cdot, \cdot)$ denoting cosine similarity, we estimate the mutual information for the mini-batch as follows.

$$I(f_\Theta(G); f_\Theta(t(G))) \to \hat{I} = \frac{1}{m}\sum_{i=1}^m \log \frac{\exp(sim(z_{i,1}, z_{i,2}))}{\sum_{i'=1, i' \neq i}^m \exp(sim(z_{i,1}, z_{i',2}))} \quad (9)$$

## 4 Related Work

GNNs for GRL is a broad field and gets a high-level review in the Sec. 1. Here, we focus on the topics that are most relevant to graph contrastive learning (GCL).

Contrastive learning (CL) [39, 59, 60, 63–65] was initially proposed to train CNNs for image representation learning and has recently achieved great success [62, 66]. GCL applies the idea of CL on GNNs. In contrast to the case of CNNs, GCL trained using GNNs posts us new fundamental challenges. An image often has multiple natural views, say by imposing different color filters and so on. Hence,

different views of an image give natural contrastive pairs for CL to train CNNs. However, graphs are more abstract and the irregularity of graph structures typically provides crucial information. Thus, designing contrastive pairs for GCL must play with irregular graph structures and thus becomes more challenging. Some works use different parts of a graph to build contrastive pairs, including nodes *v.s.* whole graphs [18, 67], nodes *v.s.* nodes [68], nodes *v.s.* subgraphs [17, 69]. Other works adopt graph data augmentations (GDA) such as edge perturbation [31] to generate contrastive pairs. Recently. GraphCL [24] gives an extensive study on different combinations of GDAs including node dropping, edge perturbation, subgraph sampling and feature masking. Extensive evaluation is required to determine good combinations. MVGRL [25] and GCA [30] leverage the domain knowledge of network science and adopt network centrality to perform GDAs. Note that none of the above methods consider optimizing augmentations. In contrast, our principle AD-GCL provides theoretical guiding principles to optimize augmentations. Very recently, JOAO [70] adopts a bi-level optimization framework sharing some high-level ideas with our adversarial training strategy but has several differences: 1) the GDA search space in JOAO is set as different types of augmentation with uniform perturbation, such as uniform edge/node dropping while we allow augmentation with non-uniform perturbation. 2) JOAO relaxes the GDA combinatorial search problem into continuous space via Jensen's inequality and adopts projected gradient descent to optimize. Ours, instead, adopts Bayesian modeling plus reparameterization tricks to optimize. The performance comparison between AD-GCL and JOAO for the tasks investigated in Sec. 5 is given in Appendix H.

Tian et al. [71] has recently proposed the InfoMin principle that shares some ideas with AD-GCL but there are several fundamental differences. Theoretically, InfoMin needs the downstream tasks to supervise the augmentation. Rephrased in our notation, the optimal augmentation $T_{IM}(G)$ given by InfoMin (called the sweet spot in [71]) needs to satisfy $I(t_{IM}(G); Y) = I(G; Y)$ and $I(t_{IM}(G); G|Y) = 0, t_{IM}(G) \sim T_{IM}(G)$, neither of which are possible without the downstream-task knowledge. Instead, our Theorem 1 provides more reasonable arguments and creatively suggests using regularization to control the tradeoff. Empirically, InfoMin is applied to CNNs while AD-GCL is applied to GNNs. AD-GCL needs to handle the above challenges due to irregular graph structures and the limited expressive power of GNNs [14, 15], which InfoMin does not consider.

## 5   Experiments and Analysis

This section is devoted to the empirical evaluation of the proposed instantiation of our AD-GCL principle. Our initial focus is on unsupervised learning which is followed by analysis of the effects of regularization. We further apply AD-GCL to transfer and semi-supervised learning. Summary of datasets and training details for specific experiments are provided in Appendix E and G respectively.

### 5.1   Unsupervised Learning

In this setting, an encoder (specifically GIN [72]) is trained with different self-supervised methods to learn graph representations, which are then evaluated by feeding these representations to make prediction for the downstream tasks. We use datasets from Open Graph Benchmark (OGB) [52], TU Dataset [73] and ZINC [74] for graph-level property classification and regression. More details regarding the experimental setting are provided in the Appendix G.

We consider two types of AD-GCL, where one is with a fixed regularization weight $\lambda_{\text{reg}} = 5$ (Eq.8), termed AD-GCL-FIX, and another is with $\lambda_{\text{reg}}$ tuned over the validation set among $\{0.1, 0.3, 0.5, 1.0, 2.0, 5.0, 10.0\}$, termed AD-GCL-OPT. AD-GCL-FIX assumes any information from the downstream task as unavailable while AD-GCL-OPT assumes the augmentation search space has some weak information from the downstream task. A full range of analysis on how $\lambda_{\text{reg}}$ impacts AD-GCL will be investigated in Sec. 5.2. We compare AD-GCL with three unsupervised/self-supervised learning baselines for graph-level tasks, which include randomly initialized untrained GIN (RU-GIN) [72], InfoGraph [18] and GraphCL [24]. Previous works [18, 24] show that they generally outperform graph kernels [75–77] and network embedding methods [33, 34, 78, 79].

We also adopt GCL with GDA based on non-adversarial edge dropping (NAD-GCL) for ablation study. NAD-GCL drops the edges of a graph uniformly at random. We consider NAD-GCL-FIX and NAD-GCL-OPT with different edge drop ratios. NAD-GCL-GCL adopts the edge drop ratio of AD-GCL-FIX at the saddle point of the optimization (Eq.8) while NAD-GCL-OPT optimally tunes the edge drop ratio over the validation datasets to match AD-GCL-OPT. We also adopt fully supervised GIN (F-GIN) to provide an anchor of the performance. We stress that all methods adopt GIN [72] as the encoder. Except F-GIN, all methods adopt a downstream *linear* classifier or regressor

| Dataset | NCI1 | PROTEINS | MUTAG | DD | COLLAB | RDT-B | RDT-M5K | IMDB-B | IMDB-M |
|---|---|---|---|---|---|---|---|---|---|
| F-GIN | 78.27 ± 1.35 | 72.39 ± 2.76 | 90.41 ± 4.61 | 74.87 ± 3.56 | 74.82 ± 0.92 | 86.79 ± 2.04 | 53.28 ± 3.17 | 71.83 ± 1.93 | 48.46 ± 2.31 |
| *Baselines* RU-GIN [72] | 62.98 ± 0.10 | 69.03 ± 0.33 | 87.61 ± 0.39 | 74.22 ± 0.30 | 63.08 ± 0.10 | 58.97 ± 0.13 | 27.52 ± 0.61 | 51.86 ± 0.33 | 32.81 ± 0.57 |
| InfoGraph [18] | 68.13 ± 0.59 | 72.57 ± 0.65 | 87.71 ± 1.77 | 75.23 ± 0.39 | 70.35 ± 0.64 | 78.79 ± 2.14 | 51.11 ± 0.55 | 71.11 ± 0.88 | 48.66 ± 0.67 |
| GraphCL [24] | 68.54 ± 0.55 | 72.86 ± 1.01 | 88.29 ± 1.31 | 74.70 ± 0.70 | 71.26 ± 0.55 | 82.63 ± 0.99 | 53.05 ± 0.40 | 70.80 ± 0.77 | 48.49 ± 0.63 |
| *AB-S* NAD-GCL-FIX | 69.23 ± 0.60 | 72.81 ± 0.71 | 88.58 ± 1.58 | 74.55 ± 0.55 | 71.56 ± 0.58 | 83.41 ± 0.66 | 52.72 ± 0.71 | 70.94 ± 0.77 | 48.33 ± 0.47 |
| NAD-GCL-OPT | 69.30 ± 0.32 | 73.18 ± 0.71 | 89.05 ± 1.06 | 74.55 ± 0.55 | 72.04 ± 0.67 | 83.74 ± 0.76 | 53.43 ± 0.26 | 71.94 ± 0.59 | 49.01 ± 0.93 |
| *Ours* AD-GCL-FIX | **69.67 ± 0.51***  | **73.59 ± 0.65** | 89.25 ± 1.45 | 74.49 ± 0.52 | **73.32 ± 0.61*** | **85.52 ± 0.79*** | 53.00 ± 0.82 | 71.57 ± 1.01 | 49.04 ± 0.53 |
| AD-GCL-OPT | **69.67 ± 0.51*** | **73.81 ± 0.46*** | **89.70 ± 1.03** | 75.10 ± 0.39 | **73.32 ± 0.61*** | **85.52 ± 0.79*** | **54.93 ± 0.43*** | **72.33 ± 0.56*** | **49.89 ± 0.66*** |

| Task | Regression (Downstream Classifier - Linear Regression + L2) | | | | Classification (Downstream Classifier - Logistic Regression + L2) | | | | |
|---|---|---|---|---|---|---|---|---|---|
| Dataset | molesol | mollipo | molfreesolv | ZINC-10K | molbace | molbbbp | molclintox | moltox21 | molsider |
| Metric | RMSE (shared) (↓) | | | MAE (↓) | ROC-AUC % (shared) (↑) | | | | |
| F-GIN | 1.173 ± 0.057 | 0.757 ± 0.018 | 2.755 ± 0.349 | 0.254 ± 0.005 | 72.97 ± 4.00 | 68.17 ± 1.48 | 88.14 ± 2.51 | 74.91 ± 0.51 | 57.60 ± 1.40 |
| *Baselines* RU-GIN [72] | 1.706 ± 0.180 | 1.075 ± 0.022 | 7.526 ± 2.119 | 0.809 ± 0.022 | 75.07 ± 2.23 | 64.48 ± 2.46 | 72.29 ± 4.15 | 71.53 ± 0.74 | 62.29 ± 1.12 |
| InfoGraph [18] | 1.344 ± 0.178 | 1.005 ± 0.023 | 10.005 ± 4.819 | 0.890 ± 0.017 | 74.74 ± 3.64 | 66.33 ± 2.79 | 64.50 ± 5.32 | 69.74 ± 0.57 | 60.54 ± 0.90 |
| GraphCL [24] | 1.272 ± 0.089 | 0.910 ± 0.016 | 7.679 ± 2.748 | 0.627 ± 0.013 | 74.32 ± 2.70 | 68.22 ± 1.89 | 74.92 ± 4.42 | 72.40 ± 1.01 | 61.76 ± 1.11 |
| *AB-S* NAD-GCL-FIX | 1.392 ± 0.065 | 0.952 ± 0.024 | 5.840 ± 0.877 | 0.609 ± 0.010 | 73.60 ± 2.73 | 66.12 ± 1.80 | 73.32 ± 3.66 | 71.65 ± 0.94 | 60.41 ± 1.48 |
| NAD-GCL-OPT | 1.242 ± 0.096 | 0.897 ± 0.022 | 5.840 ± 0.877 | 0.609 ± 0.010 | 73.69 ± 3.67 | 67.70 ± 1.78 | 74.40 ± 4.92 | 71.65 ± 0.94 | 61.14 ± 1.43 |
| *Ours* AD-GCL-FIX | **1.217 ± 0.087** | **0.842 ± 0.028*** | **5.150 ± 0.624*** | **0.578 ± 0.012*** | **76.37 ± 2.03** | 68.24 ± 1.47 | **80.77 ± 3.92** | 71.42 ± 0.73 | **63.19 ± 0.95** |
| AD-GCL-OPT | **1.136 ± 0.050*** | **0.812 ± 0.020*** | **4.145 ± 0.369*** | **0.544 ± 0.004*** | **77.27 ± 2.56** | **69.54 ± 1.92** | **80.77 ± 3.92** | **72.92 ± 0.86** | **63.19 ± 0.95** |

Table 1: Unsupervised learning performance for (TOP) biochemical and social network classification in TU datasets [73] (Averaged accuracy ± std. over 10 runs) and (BOTTOM) chemical molecules property prediction in OGB datasets [52] (mean ± std. over 10 runs). **Bold/Bold*** indicates our methods outperform baselines with $\geq 0.5 / \geq 2$ std respectively. Fully supervised (F-GIN) results are shown **only** for placing GRL methods in perspective. Ablation-study (AB-S) results do not count as baselines.

with the same hyper-parameters for fair comparison. Adopting *linear models* was suggested by [40], which explicitly attributes any performance gain/drop to the quality of learnt representations.

Tables 1 show the results for unsupervised graph level property prediction in social and chemical domains respectively. We witness the big performance gain of AD-GCL as opposed to all baselines across all the datasets. Note GraphCL utilizes extensive evaluation to select the best combination of augmentions over a broad GDA family including node-dropping, edge dropping and subgraph sampling. Our results indicate that such extensive evaluation may not be necessary while optimizing the augmentation strategy in an adversarial way is greatly beneficial.

We stress that edge dropping is not cherry picked as the search space of augmentation strategies. Other search spaces may even achieve better performance, while an extensive investigation is left for the future work.

Moreover, AD-GCL also clearly improves upon the performance against its non-adversarial counter-parts (NAD-GCL) across all the datasets, which further demonstrates stable and significant advantages of the AD-GCL principle. Essentially, the input-graph-dependent augmentation learnt by AD-GCL yields much benefit. Finally, we compare AD-GCL-FIX with AD-GCL-OPT. Interestingly, two methods achieve comparable results though AD-GCL-OPT is sometimes better. This observation implies that the AD-GCL principle may be robust to the choice of $\lambda_{\text{reg}}$ and thus motivates the analysis in the next subsection. Moreover, weak information from the downstream tasks indeed help with controlling the search space and further betters the performance. We also list the optimal $\lambda_{\text{reg}}$'s of AD-GCL-OPT for different datasets in Appendix F.1 for the purpose of comparison and reproduction.

### 5.1.1 Note on the linear downstream classifier

We find that the choice of the downstream classifier can significantly affect the evaluation of the self-supervised representations. InfoGraph [18] and GraphCL [24] adopt a non-linear SVM model as the downstream classifier. Such a non-linear model is more powerful than the linear model we adopt and thus causes some performance gap between the results showed in Table 1 (TOP) and (BOTTOM) and their original results (listed in Appendix G.2.1 as Table 8). We argue that using a non-linear SVM model as the downstream classifier is unfair, because the performance of even a randomly initialized untrained GIN (RU-GIN) is significantly improved (comparing results from Table 1 (TOP) to Table 8 ). Therefore, we argue for adopting a linear classifier protocol as suggested by [40]. That having been said, our methods (both AD-GCL-FIX and AD-GCL-OPT) still performs significantly better than baselines in most cases, even when a non-linear SVM classifer is adopted, as shown in Table 8. Several relative gains are there no matter whether the downstream classifier is a simple linear

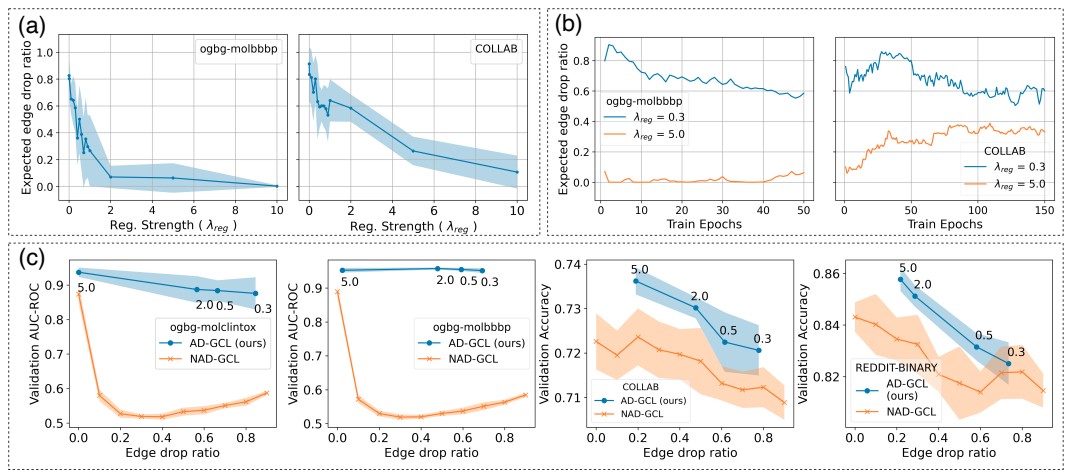

Figure 3: (a) $\lambda_{\text{reg}}$ *v.s.* expected edge drop ratio $\mathbb{E}_{\mathcal{G}}[\sum_e \omega_e / |E|]$ (measured at saddle point of Eq.8). (b) Training dynamics of expected drop ratio for $\lambda_{\text{reg}}$. (c) Validation performance for graph classification *v.s.* edge drop ratio. Compare AD-GCL and GCL with non-adversarial edge dropping. The markers on AD-GCL's performance curves show the $\lambda_{\text{reg}}$ used.

model (Tables 1) or a non-linear SVM model (Table 8). AD-GCL methods significantly outperform InfoGraph in 5 over 8 datasets and GraphCL in 6 over 8 datasets. This further provides the evidence for the effectiveness of our method. Details on the practical benefits of linear downstream models can be found in Appendix G.2.1.

## 5.2 Analysis of Regularizing the GDA Model

Here, we study how different $\lambda_{\text{reg}}$'s impact the expected edge drop ratio of AD-GCL at the saddle point of Eq.8 and further impact the model performance on the validation datasets. Due to the page limitation, we focus on classification tasks in the main text while leaving the discussion on regression tasks in the Appendix F.2. Figure 3 shows the results.

As shown in Figure 3(a), a large $\lambda_{\text{reg}}$ tends to yield a small expected edge drop ratio at the convergent point, which matches our expectation. $\lambda_{\text{reg}}$ ranging from 0.1 to 10.0 corresponds to dropping almost everything (80% edges) to nothing (<10% edges). The validation performance in Figure 3(c) is out of our expectation. We find that for classification tasks, the performance of the encoder is extremely robust to different choices of $\lambda_{\text{reg}}$'s when trained w.r.t. the AD-GCL principle, though the edge drop ratios at the saddle point are very different. However, the non-adversarial counterpart NAD-GCL is sensitive to different edge drop ratios, especially on the molecule dataset (e.g., ogbg-molclitox, ogbg-molbbbp). We actually observe the similar issue of NAD-GCL across all molecule datasets (See Appendix F.3). More interesting aspects of our results appear at the extreme cases. When $\lambda_{\text{reg}} \geq 5.0$, the convergent edge drop ratio is close to 0, which means no edge dropping, but AD-GCL still significantly outperforms naive GCL with small edge drop ratio. When $\lambda_{\text{reg}} = 0.3$, the convergent edge drop ratio is greater than 0.6, which means dropping more than half of the edges, but AD-GCL still keeps reasonable performance. We suspect that such benefit comes from the training dynamics of AD-GCL (examples as shown in Figure 3(b)). Particularly, optimizing augmentations allows for non-uniform edge-dropping probability. During the optimization procedure, AD-GCL pushes high drop probability on redundant edges while low drop probability on critical edges, which allows the encoder to differentiate redundant and critical information. This cannot be fully explained by the final convergent edge drop ratio and motivates future investigation of AD-GCL from a more in-depth theoretical perspective.

## 5.3 Transfer Learning

Next, we evaluate the GNN encoders trained by AD-GCL on transfer learning to predict chemical molecule properties and biological protein functions. We follow the setting in [17] and use the same datasets: GNNs are pre-trained on one dataset using self-supervised learning and later fine-tuned on another dataset to test out-of-distribution performance. Here, we only consider AD-GCL-FIX as AD-GCL-OPT is only expected to have better performance. We adopt baselines including no pre-trained GIN (*i.e.,* without self-supervised training on the first dataset and with only fine-tuning), InfoGraph [18], GraphCL [24], three different pre-train strategies in [17] including edge prediction,

| Pre-Train Dataset | | | | ZINC 2M | | | | | PPI-306K |
|---|---|---|---|---|---|---|---|---|---|
| Fine-Tune Dataset | BBBP | Tox21 | SIDER | ClinTox | BACE | HIV | MUV | ToxCast | PPI |
| No Pre-Train | 65.8 ± 4.5 | 74.0 ± 0.8 | 57.3 ± 1.6 | 58.0 ± 4.4 | 70.1 ± 5.4 | 75.3 ± 1.9 | 71.8 ± 2.5 | 63.4 ± 0.6 | 64.8 ± 1.0 |
| EdgePred [17] | 67.3 ± 2.4 | 76.0 ± 0.6 | 60.4 ± 0.7 | 64.1 ± 3.7 | 79.9 ± 0.9 | 76.3 ± 1.0 | 74.1 ± 2.1 | 64.1± 0.6 | 65.7 ± 1.3 |
| AttrMasking [17] | 64.3 ± 2.8 | 76.7 ± 0.4 | 61.0 ± 0.7 | 71.8 ± 4.1 | 79.3 ± 1.6 | 77.2 ± 1.1 | 74.7 ± 1.4 | 64.2 ± 0.5 | 65.2 ± 1.6 |
| ContextPred [17] | 68.0 ± 2.0 | 75.7 ± 0.7 | 60.9 ± 0.6 | 65.9 ± 3.8 | 79.6 ± 1.2 | 77.3 ± 1.0 | 75.8 ± 1.7 | 63.9 ± 0.6 | 64.4 ± 1.3 |
| InfoGraph [18] | 68.8 ± 0.8 | 75.3 ± 0.5 | 58.4 ± 0.8 | 69.9 ±3.0 | 75.9 ± 1.6 | 76.0 ± 0.7 | 75.3 ± 2.5 | 62.7 ± 0.4 | 64.1 ± 1.5 |
| GraphCL [24] | 69.68 ± 0.67 | 73.87 ± 0.66 | 60.53 ± 0.88 | 75.99 ± 2.65 | 75.38 ± 1.44 | 78.47 ± 1.22 | 69.8 ± 2.66 | 62.40 ± 0.57 | 67.88 ± 0.85 |
| AD-GCL-FIX | 70.01 ±1.07 | 76.54 ± 0.82 | **63.28 ± 0.79** | **79.78 ± 3.52** | 78.51 ± 0.80 | 78.28 ± 0.97 | 72.30 ± 1.61 | 63.07 ± 0.72 | **68.83 ± 1.26** |
| Our Ranks | 1 | 2 | 1 | 1 | 4 | 2 | 5 | 5 | 1 |

Table 2: Transfer learning performance for chemical molecules property prediction (mean ROC-AUC ± std. over 10 runs). **Bold** indicates our methods outperform baselines with $\geq 0.5$ std..

| Dataset | NCI1 | PROTEINS | DD | COLLAB | RDT-B | RDT-M5K |
|---|---|---|---|---|---|---|
| No Pre-Train | 73.72 ± 0.24 | 70.40 ± 1.54 | 73.56 ± 0.41 | 73.71± 0.27 | 86.63 ± 0.27 | 51.33 ± 0.44 |
| SS-GCN-A | 73.59 ± 0.32 | 70.29 ± 0.64 | 74.30 ± 0.81 | 74.19 ± 0.13 | 87.74 ± 0.39 | 52.01 ± 0.20 |
| GAE [20] | 74.36 ± 0.24 | 70.51 ± 0.17 | 74.54 ± 0.68 | 75.09 ± 0.19 | 87.69 ± 0.40 | 53.58 ± 0.13 |
| InfoGraph [18] | 74.86 ± 0.26 | 72.27 ± 0.40 | 75.78 ± 0.34 | 73.76 ± 0.29 | 88.66 ± 0.95 | 53.61 ± 0.31 |
| GraphCL [24] | 74.63 ± 0.25 | 74.17 ± 0.34 | 76.17 ± 1.37 | 74.23 ± 0.21 | 89.11 ± 0.19 | 52.55 ± 0.45 |
| AD-GCL-FIX | **75.18 ± 0.31** | 73.96 ± 0.47 | **77.91 ± 0.73\*** | **75.82 ± 0.26\*** | **90.10 ± 0.15\*** | 53.49 ± 0.28 |
| Our Ranks | 1 | 2 | 1 | 1 | 1 | 3 |

Table 3: Semi-supervised learning performance with 10% labels on TU datasets [73] (10-Fold Accuracy (%)± std over 5 runs). **Bold/Bold\*** indicate our methods outperform baselines with $\geq 0.5$ std/ $\geq 2$ std respectively.

node attribute masking and context prediction that utilize edge, node and subgraph context respectively. More detailed setup is given in Appendix G.

According to Table 2, AD-GCL-FIX significantly outperforms baselines in 3 out of 9 datasets and achieves a mean rank of 2.4 across these 9 datasets which is better than all baselines. Note that although AD-GCL only achieves 5th on some datasets, AD-GCL still significantly outperforms InfoGraph [18] and GraphCL [24], both of which are strong GNN self-training baselines. In contrast to InfoGraph [18] and GraphCL [24], AD-GCL achieves some performance much closer to those baselines (EdgePred, AttrMasking and ContextPred) based on domain knowledge and extensive evaluation in [17]. This is rather significant as our method utilizes only edge dropping GDA, which again shows the effectiveness of the AD-GCL principle.

## 5.4 Semi-Supervised Learning

Lastly, we evaluate AD-GCL on semi-supervised learning for graph classification on the benchmark TU datasets [73]. We follow the setting in [24]: GNNs are pre-trained on one dataset using self-supervised learning and later fine-tuned based on 10% label supervision on the same dataset. Again, we only consider AD-GCL-FIX and compare it with several baselines in [24]: 1) no pre-trained GCN, which is directly trained by the 10% labels from scratch, 2) SS-GCN-A, a baseline that introduces more labelled data by creating random augmentations and then gets trained from scratch, 3) a predictive method GAE [20] that utilizes adjacency reconstruction in the pre-training phase, and GCL methods, 4) InfoGraph [18] and 5) GraphCL [24]. Note that here we have to keep the encoder architecture same and thus AD-GCL-FIX adopts GCN as the encoder. Table 3 shows the results. AD-GCL-FIX significantly outperforms baselines in 3 out of 6 datasets and achieves a mean rank of 1.5 across these 6 datasets, which again demonstrates the strength of AD-GCL.

## 6 Conclusions

In this work we have developed a theoretically motivated, novel principle: *AD-GCL* that goes a step beyond the conventional InfoMax objective for self-supervised learning of GNNs. The optimal GNN encoders that are agnostic to the downstream tasks are the ones that capture the minimal sufficient information to identify each graph in the dataset. To achieve this goal, AD-GCL suggests to better graph contrastive learning via optimizing graph augmentations in an adversarial way. Following this principle, we developed a practical instantiation based on learnable edge dropping. We have extensively analyzed and demonstrated the benefits of AD-GCL and its instantiation with real-world datasets for graph property prediction in unsupervised, transfer and semi-supervised learning settings.

## Acknowledgments and Disclosure of Funding

We greatly thank the actionable suggestions given by reviewers and the area chair. S.S. and J.N. are supported by the National Science Foundation under contract numbers CCF-1918483 and IIS-1618690. P.L. is partly supported by the 2021 JP Morgan Faculty Award and the National Science Foundation (NSF) award HDR-2117997.

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
