# A Summary of the Appendix

In the appendix, we provide the detailed proof of the Theorem 1 (Sec. B), a review of WL tests (Sec. C), the detailed algorithmic format of our instantiation of AD-GCL (Sec. D), the summary of datasets (Sec. E), more regularization hyperparameter analysis (Sec. F), detailed experimental settings and complete evaluation results (Sec. G), computing resources (Sec. J) and discussion on broader impacts (Sec.I).

# B Proof of Theorem 1

We repeat Theorem 1 as follows.

**Theorem 2.** *Suppose the encoder $f$ is implemented by a GNN as powerful as the 1-WL test. Suppose $\mathcal{G}$ is a countable space and thus $\mathcal{G}'$ is a countable space. Then, the optimal solution $(f^*, T^*)$ to AD-GCL satisfies, letting $T'^*(G') = \mathbb{E}_{G \sim \mathbb{P}_{\mathcal{G}}}[T^*(G)|G \cong G']$,*

1. $I(f^*(t^*(G)); G \,|\, Y) \leq \min_{T \in \mathcal{T}} I(t'(G'); G') - I(t'^*(G'); Y)$, *where* $t^*(G) \sim T^*(G)$, $t'(G') \sim T'(G')$, $t'^*(G') \sim T'^*(G')$, $(G, Y) \sim \mathbb{P}_{\mathcal{G} \times \mathcal{Y}}$ *and* $(G', Y) \sim \mathbb{P}_{\mathcal{G}' \times \mathcal{Y}}$.

2. $I(f^*(G); Y) \geq I(f^*(t'^*(G')); Y) = I(t'^*(G'); Y)$, *where* $t^*(G) \sim T^*(G)$, $t'^*(G') \sim T'^*(G')$, $(G, Y) \sim \mathbb{P}_{\mathcal{G} \times \mathcal{Y}}$ *and* $(G', Y) \sim \mathbb{P}_{\mathcal{G}' \times \mathcal{Y}}$.

*Proof.* Because $\mathcal{G}$ and $\mathcal{G}'$ are countable, $P_{\mathcal{G}}$ and $P_{\mathcal{G}'}$ are defined over countable sets and thus discrete distribution. Later we may call a function $z(\cdot)$ can distinguish two graphs $G_1, G_2$ if $z(G_1) \neq z(G_2)$.

Moreover, for notational simplicity, we consider the following definition. Because $f^*$ is as powerful as the 1-WL test. Then, for any two graphs $G_1, G_2 \in \mathcal{G}$, $G_1 \cong G_2$, $f^*(G_1) = f^*(G_2)$. We may define a mapping over $\mathcal{G}'$, also denoted by $f^*$ which simply satisfies $f^*(G') :\triangleq f^*(G)$, where $G \cong G'$, and $G \in \mathcal{G}$ and $G' \in \mathcal{G}'$.

We first prove the statement 1, *i.e.,* the upper bound. We have the following inequality: Recall that $T'^*(G') = \mathbb{E}_{G \sim \mathbb{P}_{\mathcal{G}}}[T^*(G)|G \cong G']$ and $t'^*(G') \sim T'^*(G')$.

$$
\begin{aligned}
I(t'^*(G'); G') &= I(t'^*(G'); (G', Y)) - I(t'^*(G'); Y|G')] \\
&\overset{(a)}{=} I(t'^*(G'); (G', Y)) \\
&= I(t'^*(G'); Y) + I(t'^*(G'); G'|Y) \\
&\overset{(b)}{\geq} I(f^*(t'^*(G')); G'|Y) + I(t'^*(G'); Y) \quad (10)
\end{aligned}
$$

where $(a)$ is because $t'^*(G') \perp_{G'} Y$, $(b)$ is because the data processing inequality [80]. Moreover, because $f^*$ could be as powerful as the 1-WL test and thus could be injective in $\mathcal{G}'$ a.e. with respect to the measure $\mathbb{P}_{\mathcal{G}'}$. Then, for any GDA $T(\cdot)$ and $T'(G') = \mathbb{E}_{G \sim \mathbb{P}_{\mathcal{G}}}[T(G)|G \cong G']$,

$$
I(t'(G'); G') = I(f^*(t'(G')); f^*(G')) = I(f^*(t(G)); f^*(G)), \quad (11)
$$

where $t'(G') \sim T'(G')$, $t(G) \sim T(G)$. Here, the second equality is because $f^*(G) = f^*(G')$ and $T'(G') = \mathbb{E}_{G \sim \mathbb{P}_{\mathcal{G}}}[T(G)|G \cong G']$.

Since $T^* = \arg\min_{T \in \mathcal{T}} I(f(t^*(G)); f(G))$ where $t^*(G) \sim T^*(G)$ and Eq.11, we have

$$
I(t'^*(G'); G') = \arg\min_{T \in \mathcal{T}} I(t'(G'); G'), \text{ where } t'(G') \sim T'(G') = \mathbb{E}_{G \sim \mathbb{P}_{\mathcal{G}}}[T(G)|G \cong G']. \quad (12)
$$

Again, because by definition $f^* = \arg\max_f I(f(G); f(t^*(G)))$ and $f^*$ could be as powerful as the 1-WL test, its counterpart defined over $\mathcal{G}'$, i.e., $f^\star$, must be injective over $\mathcal{G}' \cap \mathrm{Supp}(\mathbb{E}_{G' \sim \mathbb{P}_{\mathcal{G}'}}[T'^*(G')])$ a.e. with respect to the measure $\mathbb{P}_{\mathcal{G}'}$ to achieve such mutual information maximization. Here, $\mathrm{Supp}(\mu)$ defines the set where $\mu$ has non-zero measure. Because of the definition of $T'^*(G') = \mathbb{E}_{G \sim \mathbb{P}_{\mathcal{G}}}[T^*(G)|G \cong G']$,

$$
\mathcal{G}' \cap \mathrm{Supp}(\mathbb{E}_{G' \sim \mathbb{P}_{\mathcal{G}'}}[T'^*(G')]) = \mathcal{G}' \cap \mathrm{Supp}(\mathbb{E}_{G \sim \mathbb{P}_{\mathcal{G}}}[T^*(G)]).
$$

Therefore, $f^*$ is a.e. injective over $\mathcal{G}' \cap \text{Supp}(\mathbb{E}_{G\sim\mathbb{P}_\mathcal{G}}[T^*(G)])$ and thus

$$I(f^*(t'^*(G')); G'|Y) = I(f^*(t^*(G)); G'|Y), \tag{13}$$

Moreover, as $f^*$ cannot cannot distinguish more graphs in $\mathcal{G}$ than $\mathcal{G}'$ as the power of $f^*$ is limited by 1-WL test, thus,

$$I(f^*(t^*(G)); G'|Y) = I(f^*(t^*(G)); G|Y). \tag{14}$$

Plugging Eqs.12,13,14 into Eq.10, we achieve

$$I(f^*(t^*(G)); G|Y) \leq \underset{T\in\mathcal{T}}{\arg\min}\, I(t'(G'); G') - I(t'^*(G'); Y)$$

where $t'(G') \sim T'(G') = \mathbb{E}_{G\sim\mathbb{P}_\mathcal{G}}[T(G)|G \cong G']$ and $t'^*(G') \sim T'^*(G') = \mathbb{E}_{G\sim\mathbb{P}_\mathcal{G}}[T^*(G)|G \cong G']$, which gives us the statement 1, which is the upper bound.

We next prove the statement 2, *i.e.,* the lower bound. Recall $(T^*, f^*)$ is the optimal solution to Eq.6 and $t^*(\cdot)$ denotes samples from $T^*(\cdot)$.

Again, because $f^* = \arg\max_f I(f(G); f(t^*(G)))$, $f^*$ must be injective on $\mathcal{G}' \cap \text{Supp}(\mathbb{E}_{G'\sim\mathbb{P}_{\mathcal{G}'}}[T'^*(G')])$ a.e. with respect to the measure $\mathbb{P}_{\mathcal{G}'}$. Given $t'^*(G'), t'^*(G') \to f^*(t'^*(G'))$ is an injective deterministic mapping. Therefore, for any random variable $Q$,

$$I(f^*(t'^*(G')); Q) = I(t'^*(G'); Q), \quad \text{where } G' \sim \mathbb{P}_{\mathcal{G}'}, t'^*(G') \sim T'^*(G').$$

Of course, we may set $Q = Y$. So,

$$I(f^*(t'^*(G')); Y) = I(t'^*(G'); Y), \quad \text{where } (G', Y) \sim \mathbb{P}_{\mathcal{G}'\times\mathcal{Y}}, t'^*(G') \sim T'^*(G'). \tag{15}$$

Because of the data processing inequality [80] and $T'^*(G') = \mathbb{E}_{G\sim\mathbb{P}_\mathcal{G}}[T^*(G)|G \cong G']$, we further have

$$I(f^*(t^*(G)); Y) \geq I(f^*(t'^*(G')); Y), \tag{16}$$

where $(G', Y) \sim \mathbb{P}_{\mathcal{G}'\times\mathcal{Y}}, (G, Y) \sim \mathbb{P}_{\mathcal{G}\times\mathcal{Y}}, t'^*(G') \sim T'^*(G'), t^*(G) \sim T^*(G)$.

Further because of the data processing inequality [80],

$$I(f^*(G); Y) \geq I(f^*(t^*(G)); Y). \tag{17}$$

Combining Eqs.15, 16, 17, we have

$$I(f^*(G); Y) \geq I(f^*(t^*(G)); Y) \geq I(f^*(t'^*(G')); Y) = I(t'^*(G'); Y),$$

which concludes the proof of the lower bound.

$\square$

## C   A Brief Review of the Weisfeiler-Lehman (WL) Test

Two graphs $G_1$ and $G_2$ are called to be isomorphic if there is a mapping between the nodes of the graphs such that their adjacencies are preserved. For a general class of graphs, without the knowledge of the mapping, determining if $G_1$ and $G_2$ are indeed isomorphic is challenging and there has been no known polynomial time algorithms utill now [81]. The best algorithm till now has complexity $2^{O(\log n)^3}$ where $n$ is the size of the graphs of interest [82].

The family of Weisfeiler-Lehman tests [51] (specifically the 1-WL test) offers a very efficient way perform graph isomorphism testing by generating canonical forms of graphs. Specifically, the 1-WL test follows an iterative color refinement algorithm. Let, graph $G = (V, E)$ and let $C : V \to \mathcal{C}$ denote a coloring function that assigns each vertex $v \in V$ a color $C_v$. Nodes with different features are associated with different colors. These colors constitute the initial colors $C_0$ of the algorithm i.e. $C_{0,v} = C_v$ for every vertex $v \in V$. Now, for each vertex $v$ and each iteration $i$, the algorithm creates a new set of colors from the color $C_{i-1,v}$ and the colors $C_{i-1,u}$ of every vertex $u$ that is adjacent to $v$. This multi-set of colors is then mapped to a new color (say using a unique hash). Basically, the color refinement follows

$$C_{i,v} \leftarrow \text{Hash}(C_{i-1,v}, \{C_{i-1,u}|u\in\mathcal{N}_v\}), \tag{18}$$

where the above Hash function is an injective mapping. This iteration goes on until when the list of colors stabilises, i.e. at some iteration $N$, no new colors are created. The final set of colors serves as the the the canonical form of a graph.

Intuitively, if the canonical forms obtained by 1-WL test for two graphs are different, then the graphs are surely not isomorphic. But, it is possible for two non-isomorphic graphs to share a the same 1-WL canonical form. Though the 1-WL test can test most of the non-isomorphic graphs, it will fail in some corner cases. For example, it cannot distinguish regular graphs with the same node degrees and of the same sizes.

As GNNs share the same iterative procedure as the 1-WL test by comparing Eq. 18 and Eq. 1, GNNs are proved to be at most as powerful as the 1-WL test to distinguish isormorphic graphs [14, 15]. However, GNNs with proper design may achieve the power of the 1-WL test [14] and thus the assumption in Theorem 1 is reasonable.

# D    The Training Algorithm for the Instantiation of AD-GCL

Algorithm 1 describes the self-supervised training algorithm for AD-GCL with learnable edge-dropping GDA. Note that augmenter $T_\Phi(\cdot)$ with parameters $\Phi$ is implemented as a GNN followed by an MLP to obtain the Bernoulli weights $\omega_e$.

# E    Summary of Datasets

A wide variety of datasets from different domains for a range of graph property prediction tasks are used for our experiments. Here, we summarize and point out the specific experiment setting for which they are used.

- Table 4 shows the datasets for chemical molecular property prediction which are from Open Graph Benchmark (OGB) [52] and ZINC-10K [74]. These are used in the unsupervised learning setting for both classification and regression tasks. We are the first one to considering using regression tasks and the corresponding datasets in the evaluation of self-supervised GNN.

- Table 5 shows the datasets which contain biochemical and social networks. These are taken from the TU Benchmark Datasets [73]. We use them for graph classification tasks in both unsupervised and semi-supervised learning settings.

- Table 6 shows the datasets consisting of biological interactions and chemical molecules from [17]. These datasets are used for graph classification in the transfer learning setting.

# F    Complete Results on Regularization Analysis

The main hyper-parameter for our method AD-GCL is the regularization strength $\lambda_{\text{reg}}$. Detailed sensitivity analysis is provided in Figures 3, 5 and 6. For the method AD-GCL-OPT, we tune $\lambda_{\text{reg}}$ over the validation set among $\{0.1, 0.3, 0.5, 1.0, 2.0, 5.0, 10.0\}$. For the ablation study, i.e. NAD-GCL-OPT the random edge drop ratio is tuned over the validation set among $\{0.0, 0.1, 0.2, 0.3, 0.4, 0.5, 0.6, 0.7, 0.8, 0.9\}$.

## F.1    Optimal regularization strength values

Table 7 shows the optimal $\lambda_{\text{reg}}$ on the validation set that are used to report test performance in Tables 1 (both TOP and BOTTOM).

## F.2    Effects of regularization on regression tasks

Subplots in the topmost row of Figure 5 shows the validation performance for different $\lambda_{\text{reg}}$'s in AD-GCL and random edge drop ratios in NAD-GCL for regression tasks. These observations show an interesting phenomenon that is different from what we observe in classification tasks: for AD-GCL, small $\lambda_{\text{reg}}$ (which in-turn lead to large expected edge drop ratio) results in better performance. A

**Algorithm 1:** Training Learnable Edge-Dropping GDA under AD-GCL principle.

**Input:** Data $\{G_m \sim \mathcal{G} \mid m = 1, 2 \ldots M\}$;

       Encoder $f_\Theta(\cdot)$; Augmenter $T_\Phi(\cdot)$; Projection Head $g_\Psi(\cdot)$; Cosine Similarity $sim(\cdot)$

**Hyper-Params:** Edge-Dropping Regularization Strength $\lambda_{\text{reg}}$; learning rates $\alpha, \beta$

**Output:** Trained Encoder $f_\Theta(\cdot)$

1 **begin**

2    **for** *number of training epochs* **do**

3       **for** *sampled minibatch* $\{G_n = (V_n, E_n) : n = 1, 2 \ldots N\}$ **do**

4          **for** *n = 1 to N* **do**

5             $h_{1,n} = f_\Theta(G_n)$

6             $z_{1,n} = g_\Psi(h_{1,n})$

7             $t(G_n) \sim T_\Phi(G_n)$

8             **set** $p_e, \forall e \in E_n$ **from** $t(G_n)$

9             $\mathcal{R}_n = \sum_{e \in E_n} p_e / |E_n|$

10            $h_{2,n} = f_\Theta(t(G_n))$;

11            $z_{1,n} = g_\Psi(h_{2,n})$

12          **end**

13          **define** $\mathcal{L}_n = -\log \frac{\exp(sim(z_{1,n}, z_{2,n}))}{\sum_{n'=1, n' \neq n}^{N} \exp(sim(z_{1,n}, z_{2,n'}))}$

           `/* calculate NCE loss for minibatch                        */`

14          $\mathcal{L} = \frac{1}{N} \sum_{n=1}^{N} \mathcal{L}_n$

           `/* calculate regularization term for minibatch              */`

15          $\mathcal{R} = \frac{1}{N} \sum_{n=1}^{N} \mathcal{R}_n$

           `/* update augmenter params via gradient ascent              */`

16          $\Phi \leftarrow \Phi + \alpha \nabla_\Phi (\mathcal{L} - \lambda_{\text{reg}} * \mathcal{R})$

           `/* update enocder & projection head`

             `params via gradient descent                            */`

17          $\Theta \leftarrow \Theta - \beta \nabla_\Theta(\mathcal{L}); \quad \Psi \leftarrow \Psi - \beta \nabla_\Psi(\mathcal{L})$

18       **end**

19    **end**

20    **return** *Encoder* $f_\Theta(\cdot)$

21 **end**

---

similar trend can be observed even for NAD-GCL, where large random edge drop ratios results in better performance. However, AD-GCL is still uniformly better that NAD-GCL in that regard. We reason that, regression tasks (different from classification tasks) are more sensitive to node level information rather than structural fingerprints and thus, the edge dropping GDA family might not be the most apt GDA family. Modelling different learnable GDA families is left for future work and these observations motivate such steps.

### F.3 Effects of regularization on edge-drop ratio as complete results in Figure 3 setting.

Figure 4 shows how different regularization strengths ($\lambda_{\text{reg}}$) affects the expected edge drop ratio for multiple datasets. These results further provides us evidence that indeed, $\lambda_{\text{reg}}$ and the expected edge drop ratio are inversely related in accordance with our design objective and thus provides us with a way of controlling the space of augmentations for our learnable edge dropping GDA.

Figure 6 shows the complete validation set performance for different edge drop ratios. AD-GCL is compared to a non-adversarial random edge dropping GCL (NAD-GCL). We choose $\lambda_{\text{reg}}$'s that result

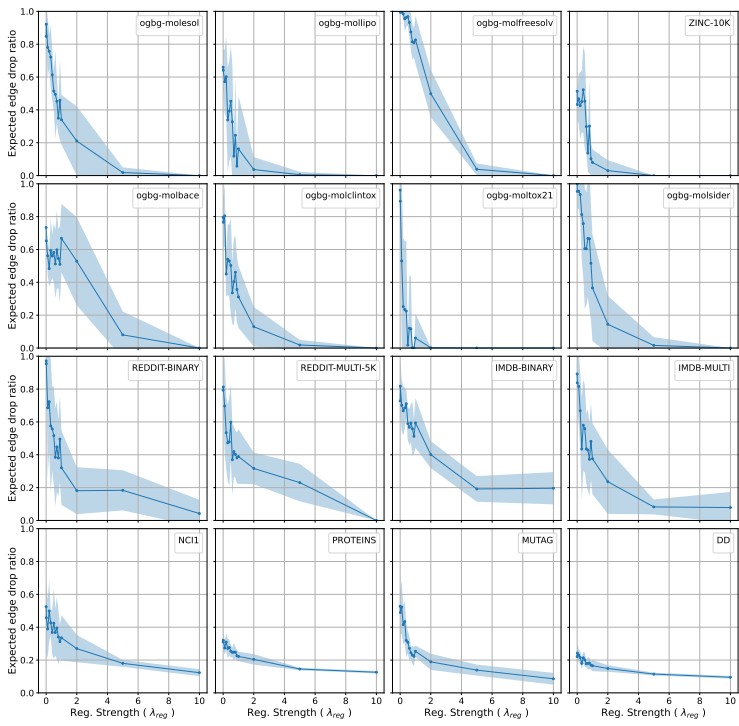

Figure 4: $\lambda_{\text{reg}}$ *v.s.* expected edge drop ratio $\mathbb{E}_{\mathcal{G}}[\sum_e \omega_e/|E|]$ (measured at saddle point of Eq.8).

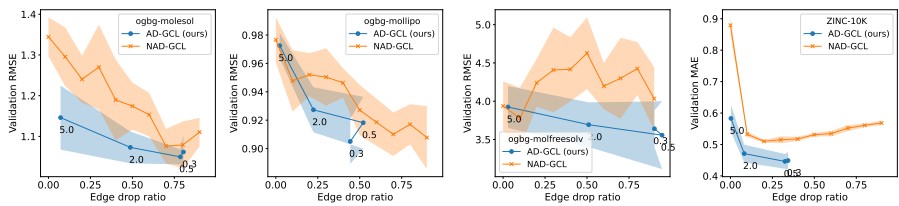

Figure 5: Validation performance for graph regression *v.s.* edge drop ratio. Comparing AD-GCL and GCL with non-adversarial random edge dropping. The markers on AD-GCL's performance curves show the $\lambda_{\text{reg}}$ used. Note here that lower validation metric is better.

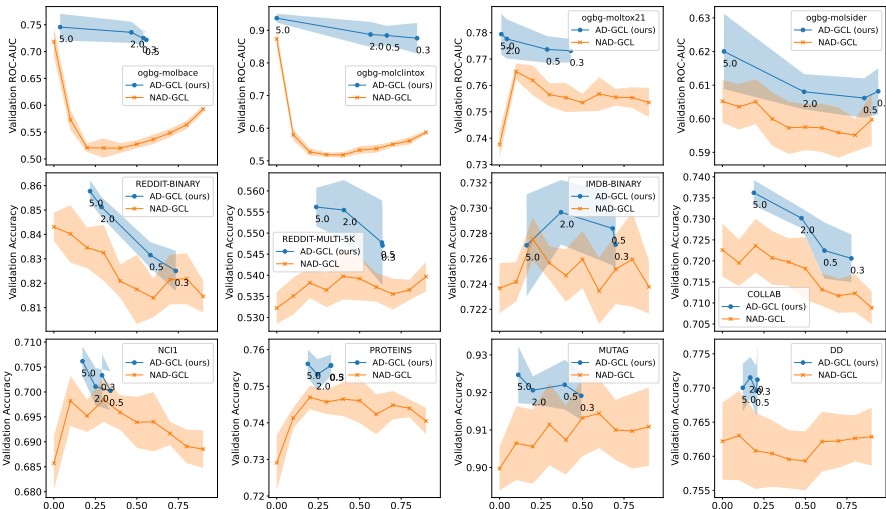

Figure 6: Validation performance for graph classification *v.s.* edge drop ratio. Comparing AD-GCL and GCL with non-adversarial random edge dropping. The markers on AD-GCL's performance curves show the $\lambda_{\text{reg}}$ used. Note here that higher validation metric is better.

| Name | #Graphs | Avg #Nodes | Avg #Edges | #Tasks | Task Type | Metric |
|---|---|---|---|---|---|---|
| ogbg-molesol | 1,128 | 13.3 | 13.7 | 1 | Regression | RMSE |
| ogbg-mollipo | 4,200 | 27.0 | 29.5 | 1 | Regression | RMSE |
| ogbg-molfreesolv | 642 | 8.7 | 8.4 | 1 | Regression | RMSE |
| ogbg-molbace | 1,513 | 34.1 | 36.9 | 1 | Binary Class. | ROC-AUC |
| ogbg-molbbbp | 2,039 | 24.1 | 26.0 | 1 | Binary Class. | ROC-AUC |
| ogbg-molclintox | 1,477 | 26.2 | 27.9 | 2 | Binary Class. | ROC-AUC |
| ogbg-moltox21 | 7,831 | 18.6 | 19.3 | 12 | Binary Class. | ROC-AUC |
| ogbg-molsider | 1,427 | 33.6 | 35.4 | 27 | Binary Class. | ROC-AUC |
| ZINC-10K | 12,000 | 23.16 | 49.83 | 1 | Regression | MAE |

Table 4: Summary of chemical molecular properties datasets used for unsupervised learning experiments. Datasets obtained from OGB [52] and [74]

| Dataset | #Graphs | Avg. #Nodes | Avg. #Edges | #Classes |
|---|---|---|---|---|
| Biochemical Molecules | | | | |
| NCI1 | 4,110 | 29.87 | 32.30 | 2 |
| PROTEINS | 1,113 | 39.06 | 72.82 | 2 |
| MUTAG | 188 | 17.93 | 19.79 | 2 |
| DD | 1,178 | 284.32 | 715.66 | 2 |
| Social Networks | | | | |
| COLLAB | 5,000 | 74.5 | 2457.78 | 3 |
| REDDIT-BINARY | 2,000 | 429.6 | 497.75 | 2 |
| REDDIT-MULTI-5K | 4,999 | 508.8 | 594.87 | 5 |
| IMDB-BINARY | 1,000 | 19.8 | 96.53 | 2 |
| IMDB-MULTI | 1,500 | 13.0 | 65.94 | 3 |

Table 5: Summary of biochemical and social networks from TU Benchmark Dataset [73] used for unsupervised and semi-supervised learning experiments. The evaluation metric for all these datasets is Accuracy.

in an expected edge drop ratio (measured at saddle point of Eq. 8) value to match the random drop ratio used for NAD-GCL.

Figure 7 further provides additional plots of the training dynamics of expected edge drop ratio for different $\lambda_{\text{reg}}$ values.

| Dataset | Utilization | #Graphs | Avg. #Nodes | Avg. Degree |
|---|---|---|---|---|
| Protein-Protein Interaction Networks | | | | |
| PPI-306K | Pre-Training | 306,925 | 39.82 | 729.62 |
| PPI | Finetuning | 88,000 | 49.35 | 890.77 |
| Chemical Molecules | | | | |
| ZINC-2M | Pre-Training | 2,000,000 | 26.62 | 57.72 |
| BBBP | Finetuning | 2,039 | 24.06 | 51.90 |
| Tox21 | Finetuning | 7,831 | 18.57 | 38.58 |
| SIDER | Finetuning | 1,427 | 33.64 | 70.71 |
| ClinTox | Finetuning | 1,477 | 26.15 | 55.76 |
| BACE | Finetuning | 1,513 | 34.08 | 73.71 |
| HIV | Finetuning | 41,127 | 25.51 | 54.93 |
| MUV | Finetuning | 93,087 | 24.23 | 52.55 |
| ToxCast | Finetuning | 8,576 | 18.78 | 38.52 |

Table 6: Summary of biological interaction and chemical molecule datasets from [17]. Used for graph classification in transfer learning experiments. The evaluation metric is ROC-AUC.

| | ogbg-molesol | ogbg-mollipo | ogbg-molfreesolv | ZINC-10K | ogbg-molbace | ogbg-molbbbp | ogbg-molclintox | ogbg-moltox21 | ogbg-molsider |
|---|---|---|---|---|---|---|---|---|---|
| AD-GCL-OPT | 0.4 | 0.1 | 0.3 | 0.8 | 10.0 | 10.0 | 5.0 | 10.0 | 5.0 |
| | COLLAB | RDT-B | RDT-M5K | IMDB-B | IMDB-M | NCI1 | PROTEINS | MUTAG | DD |
| AD-GCL-OPT | 5.0 | 5.0 | 10.0 | 2.0 | 10.0 | 5.0 | 1.0 | 10.0 | 10.0 |

Table 7: Optimal $\lambda_{\mathrm{reg}}$ for AD-GCL on validation set that are used for reporting test performance in Tables 1 (TOP) and (BOTTOM).

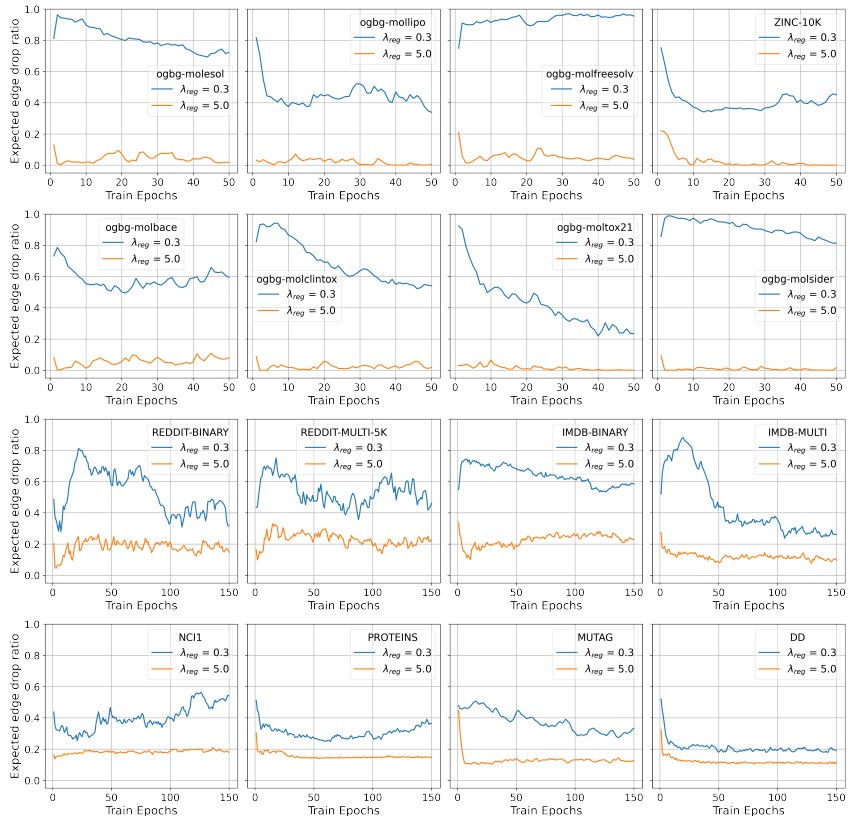

Figure 7: Training dynamics of expected edge drop ratio for $\lambda_{\mathrm{reg}}$.

# G   Experimental Settings and Complete Evaluation Results

In this section, we provide the detailed experimental settings and additional experimental evaluation results for unsupervised, transfer and semi-supervised learning experiments we conducted (Section 5). In addition we also provide details of the motivating experiment (Figure 2 in main text).

## G.1   Motivating Experiment (Figure 2)

The aim of this experiment is to show that having GNNs that can maximize mutual information between the input graph and its representation is insufficient to guarantee their performance in the downstream tasks, because redundant information may still maximize mutual information but may degenerate the performance. To show this phenomenon, we perform two case studies: (1) a GNN is trained following the vanilla GCL (InfoMax) objective and (2) a GNN is trained following the vanilla GCL (InfoMax) objective while simultaneously a linear classifier that tasks the graph representations output by the GNN encoder is trained with random labels. These two GNNs have exactly the same architectures, hyperparametes and initialization. Specifically, the GNN architecture is GIN [72], with embedding dimension of 32, 5 layers with no skip connections and a dropout of 0.0.

Both GNN encoders are trained as above. In the first step of the evaluation, we want to test whether these GNNs keep mutual information maximization. For all graphs in the ogbg-molbace dataset, either one of the GNN provides a set of graph representations. For each GNN, we compare all its

output graph representations. We find that, the output representations of every two graphs have difference that is greater than a digit accuracy. This implies that either one of the GNN keeps an one-to-one correspondance between the graphs in the dataset and their representations, which guarantees mutual information maximization.

We further compare these two GNNs encoders in the downstream task by using true labels. We impose two linear classifiers on the output representations of the above two GNN encoders to predict the true labels. The two linear classifiers have exactly the same architecture, hyperparametes and initialization. Specifically, a simple logistic classifier implemented using sklearn [83] is used with L2 regularization. The L2 strength is tuned using validation set. For the dataset ogbg-molbace, we follow the default train/val/test splits that are given by the original authors of OGB [52]. Note that, during the evaluation stage, the GNN encoders are fixed while the linear classifiers get trained. The evaluation performance is the curves as illustrated in Figure 2.

### G.2   Unsupervised Learning

**Evaluation protocol.**   In this setting, all methods are first trained with the corresponding self-supervised objective and then evaluated with a linear classifier/regressor. We follow [62] and adopt a linear evaluation protocol. Specifically, once the encoder provides representations, a Ridge regressor (+ L2) and Logistic (+ L2) classifier is trained on top and evaluated for regression and classification tasks respectively. Both methods are implemented using sklearn [83] and uses LBFGS [84] or LibLinear [85] solvers . Finally, the lone hyper-parameter of the downstream linear model i.e. L2 regularization strength is grid searched among $\{0.001, 0.01, 0.1, 1, 10, 100, 1000\}$ on the validation set for every single representation evaluation.

For the Open Graph Benchmark Datasets (ogbg-mol*), we directly use the processed data in Pytorch Geometric format which is available online [3]. The processed data includes train/val/test that follow a scaffolding split. More details are present in the OGB paper [52]. Additionally, we make use of the evaluators written by authors for standardizing the evaluation. The evaluation metric varies depending on the task at hand. For regression tasks it is RMSE (root mean square error) and for classification it is ROC-AUC (%).

For the ZINC-10K dataset [74], we use the processed data in Pytorch Geometric format that is made available online[4] by the authors. We use the same train/val/test splits that are provided. We follow the authors and adopt MAE (mean absolute error) as the test metric.

For the TU Datasets [73], we obtain the data from Pytorch Geometric Library [5] and follow the conventional 10-Fold evaluation. Following standard protocol, we adopt Accuracy (%) as the test metric.

All our experiments are performed 10 times with different random seeds and we report mean and standard deviation of the corresponding test metric for each dataset.

**Other hyper-parameters.**   The encoder used for ours and baselines is GIN [72]. The encoder is fixed and not tuned while performing self-supervised learning (i.e. embedding dimension, number of layers, pooling type) for all the methods to keep the comparison fair. The reasoning is that any performance difference we witness should only be attributed to the self-supervised objective and not to the encoder design. Details of encoder for specific datasets.

- OBG - *emb dim = 300, num gnn layers = 5, pooling = add, skip connections = None, dropout = 0.5, batch size = 32*
- ZINC-10K - *emb dim = 100, num gnn layers = 5, pooling = add, skip connections = None, dropout = 0.5, batch size = 64*
- TU Datasets - *emb dim = 32, num gnn layers = 5, pooling = add, skip connections = None, dropout = 0.5, batch size = 32*

The optimization of AD-GCL is performed using Adam and the learning rates for the encoder and the augmenter in AD-GCL are tuned among $\{0.01, 0.005, 0.001\}$. We find that asymmetric learning

---

[3]`https://ogb.stanford.edu/docs/graphprop/`
[4]`https://github.com/graphdeeplearning/benchmarking-gnns/tree/master/data`
[5]`https://pytorch-geometric.readthedocs.io/en/latest/modules/datasets.html`

rates for encoder and augmenter tend to make the training non-stable. Thus, for stability we adopt a learning rate of 0.001 for all the datasets and experiments. The number of training epochs are chosen among $\{20, 50, 80, 100, 150\}$ using the validation set.

### G.2.1 Unsupervised learning with non linear downstream classifier

| | NCI1 | PROTEINS | DD | MUTAG | COLLAB | RDT-B | RDT-M5K | IMDB-B |
|---|---|---|---|---|---|---|---|---|
| RU-GIN | $65.40 \pm 0.17$ | $72.73 \pm 0.51$ | $75.67 \pm 0.29$ | $87.39 \pm 1.09$ | $65.29 \pm 0.16$ | $76.86 \pm 0.25$ | $48.48 \pm 0.28$ | $69.37 \pm 0.37$ |
| InfoGraph | $76.20 \pm 1.06$ | $74.44 \pm 0.31$ | $72.85 \pm 1.78$ | $89.01 \pm 1.13$ | $70.65 \pm 1.13$ | $82.50 \pm 1.42$ | $53.46 \pm 1.03$ | $73.03 \pm 0.87$ |
| GraphCL | $77.87 \pm 0.41$ | $74.39 \pm 0.45$ | $78.62 \pm 0.40$ | $86.80 \pm 1.34$ | $71.36 \pm 1.15$ | $89.53 \pm 0.84$ | $55.99 \pm 0.28$ | $71.14 \pm 0.44$ |
| AD-GCL-FIX | $75.77 \pm 0.50$ | $\mathbf{75.04 \pm 0.48}$ | $75.38 \pm 0.41$ | $88.62 \pm 1.27$ | $74.79 \pm 0.41^\star$ | $92.06 \pm 0.42^\star$ | $\mathbf{56.24 \pm 0.39}$ | $71.49 \pm 0.98$ |
| AD-GCL-OPT | $75.86 \pm 0.62$ | $\mathbf{75.04 \pm 0.48}$ | $75.73 \pm 0.51$ | $88.62 \pm 1.27$ | $\mathbf{74.89 \pm 0.90^\star}$ | $\mathbf{92.35 \pm 0.42^\star}$ | $\mathbf{56.24 \pm 0.39}$ | $71.49 \pm 0.98$ |

Table 8: Unsupervised Learning results on TU Datasets using a non-linear SVM classifier as done in GraphCL [24]. Averaged Accuracy (%) $\pm$ std. over 10 runs. This is different from the linear classifier used to show results in Tables 1 (TOP) and (BOTTOM).

In our evaluation, we also observe several further benefits of using a downstream linear model in practice, would like to list them here. First, linear classifiers are much faster to train/converge in practice, especially for the large-scaled datasets or large embedding dimensions, which is good for practical usage. We observe that non-linear SVM classifiers induce a rather slow convergence, when applying to those several OGB datasets where the embedding dimensions are 300 (Table 1 bottom). Second, compared to the linear model, the non-liner SVM may introduce additional hyper-parameters which not only need further effort to be tuned but also weaken the effect of the self-training procedure on the downstream performance.

### G.3 Transfer Learning

**Evaluation protocol.** We follow the same evaluation protocol as done in [17]. In this setting, self-supervised methods are trained on the pre-train dataset and later used to be test regarding transferability. In the testing procedure, the models are fine-tuned on multiple datasets and evaluated by the labels of these datasets. We adopt the GIN encoder used in [17] with the same settings for fair comparison. All reported values for baseline methods are taken directly from [17] and [24]. For the fine-tuning, the encoder has an additional linear graph prediction layer on top which is used to map the representations to the task labels. This is trained end-to-end using gradient descent (Adam).

**Hyper-parameters.** Due to the large pre-train dataset size and multiple fine-tune datasets finding optimal $\lambda_{\mathrm{reg}}$ for each of them can become time consuming. Instead we use a fixed $\lambda_{\mathrm{reg}} = 5.0$ as it provides reasonable performance. The learning rate is also fixed to 0.001 and is symmetric for both the encoder and augmenter during self-supervision on the pre-train dataset. The number of training epochs for pre-training is chosen among $\{20, 50, 80, 100\}$ based on the validation performance on the fine-tune datasets. The same learning setting for fine-tuning is used by following [24].

### G.4 Semi-supervised Learning

**Evaluation protocol.** We follow the protocol as mentioned in [24]. In this setting, the self-supervised methods are pre-trained and later fine-tuned with 10% true label supervision on the same dataset. The representations generated by the methods are finally evaluated using 10-fold evaluation. All reported values for baseline methods are taken directly from [24]. For fine-tuning, the encoder has an additional linear graph prediction layer on top which is used to map the representations to the task labels. This is trained end-to-end by using gradient descent (Adam).

**Hyper-parameters.** For the pre-training our model, a fixed $\lambda_{\mathrm{reg}} = 5.0$ and learning rate of 0.001 for both encoder and augmenter is used. The epochs are selected among $\{20, 50, 80, 100\}$ and finally for fine-tuning with 10% label supervision, default parameters from [24] are used.

## H  Comparison of AD-GCL and JOAO

We first clarify the different mechanisms that JOAO [70] and AD-GCL adopt. JOAO selects augmentation families from a pool $\mathcal{A}$ = NodeDrop, Subgraph,EdgePert, AttrMask,Identical and defines a

| Dataset | NCI1 | PROTEINS | DD | MUTAG | COLLAB | RDT-B | RDT-M5K | IMDB-B |
|---|---|---|---|---|---|---|---|---|
| JOAO | 78.07±0.47 | 74.55±0.41 | 77.32±0.54 | 87.35±1.02 | 69.50±0.36 | 85.29±1.35 | 55.74±0.63 | 70.21±3.08 |
| JOAOv2 | 78.36±0.53 | 74.07±1.10 | 77.40±1.15 | 87.67±0.79 | 69.33±0.34 | 86.42±1.45 | 56.03±0.27 | 70.83±0.25 |
| AD-GCL-FIX | 75.77±0.50 | 75.04±0.48 | 75.38±0.41 | 88.62±1.27 | 74.79±0.41 | 92.06±0.42 | 56.24±0.39 | 71.49±0.98 |

Table 9: Unsupervised learning showing Averaged Accuracy (%) ± std. with a non linear SVM downstream classifier and same standard setup as used in [70]. The results for JOAO and JOAOv2 are taken from [70].

| Dataset | NCI1 | PROTEINS | DD | MUTAG | COLLAB | RDT-B | RDT-M5K | IMDB-B |
|---|---|---|---|---|---|---|---|---|
| JOAOv2 (FIX-gamma=0.1) | 72.99±0.75 | 71.25±0.85 | 66.91±1.75 | 85.20±1.64 | 70.40±2.21 | 78.35±1.38 | 45.57±2.86 | 71.60±0.86 |
| AD-GCL-FIX | 69.67±0.51 | 73.59±0.65 | 74.49±0.52 | 89.25±1.45 | 73.32±0.61 | 85.52±0.79 | 53.00±0.82 | 71.57±1.01 |

Table 10: Unsupervised learning showing Averaged Accuracy (%) ± std. with a linear downstream classifier. JOAOv2 results using linear evaluation is obtained by us using code provided by the authors.

uniform prior on them for their inner optimization over all possible augmentation family pairs. (See Section 3.2 and See Eq. 7,8 in [70]). An important distinction is that JOAO still adopts uniformly random augmentations and the inner optimization only searches over different pairs of uniform augmentations. Whereas, AD-GCL adopts non-uniformly random augmentations, which essentially corresponds to a much larger search space.

Complexity wise, JOAO is more expensive than AD-GCL as, they utilize projected gradient descent to fully optimize the inner optimization step over all possible augmentations $\mathcal{A}$. This is a factor k more expensive than AD-GCL. The factor k in JOAO is currently $|\mathcal{A}|^2 = 4^2 = 16$. This makes it slow to train while still having a restricted search space compared to AD-GCL which on the other hand is both faster and looks at a larger search space for a given augmentation family. In our experiments on a single GPU, JOAO took 3.2 hrs for training on COLLAB whereas AD-GCL only took 14.4 mins (0.24 hrs).

Moreover, we derive the min-max principle in a more principled way by illustrating its connection to graph information bottleneck (Theorem 1), which explains the fundamental reason and benefits of optimizing graph augmentation strategies.

### H.1    Experimental Comparison

We provide comparison between JOAO and AD-GCL in unsupervised learning setting with the standard non-linear downstream classifier setting in Table 9. This is done following [70] for fair comparison. Now, we provide the comparison between JOAO and AD-GCL using a linear evaluation protocol for unsupervised setting in Table 10. Specifically, a linear SVM head is used for evaluating the representations learned by the 2 methods for the downstream task. The regularization hyper-parameter of the linear svm is grid-searched among 0.001, 0.01,0.1,1,10,100,1000. We re-run the code provided by authors of JOAO (available at `https://github.com/Shen-Lab/GraphCL_Automated`) with the default parameters for 5 times each with different seeds. The only change done is to the embedding evaluation code to include linear svm as the final prediction head. For all the TU datasets used here, standard 10-Fold evaluation is used to report classification accuracy (%).

The results in the above table further indicate that AD-GCL performs better than JOAO in 6 of the 8 TU benchmark datasets. The gap in performance is even more clear compared to the non-linear evaluation setting as shown previously in Table 9. Again, we reiterate that the improved performance gains are due to AD-GCL's search of non-uniformly random augmentations.

More comparison on transfer learning and semi-supervised learning is put in Table 11 and Table 11 respectively, where the experimental settings follow Sec. 5. For transfer learning, AD-GCL outperforms JOAO in 7 among 9 datasets, JOAOv2 in 5 among 9 datasets. For semi-supervised learning, AD-GCL outperforms both of them in all 6 datasets.

| Fine-Tune Dataset | BBBP | Tox21 | SIDER | ClinTox | BACE | HIV | MUV | ToxCast | PPI |
|---|---|---|---|---|---|---|---|---|---|
| JOAO | 70.22±0.98 | 74.98±0.29 | 59.97±0.79 | 81.32±2.49 | 77.34±0.48 | 76.73±1.23 | 71.66±1.43 | 62.94±0.48 | 64.43±1.38 |
| JOAOv2 | 71.39±0.92 | 74.27±0.62 | 60.49±0.74 | 80.97±1.64 | 75.49±1.27 | 77.51±1.17 | 73.67±1.00 | 63.16±0.45 | 63.94±1.59 |
| AD-GCL-FIX | 70.01±1.07 | 76.54±0.82 | 63.28±0.79 | 79.78±3.52 | 78.51±0.80 | 78.28±0.97 | 72.30±1.61 | 63.07±0.72 | 68.83±1.26 |

Table 11: Transfer learning results showing mean ROC-AUC $\pm$ std. Pre-Training done using ZINC 2M (used for first 8 fine-tune datasets) and PPI-306K (for the last PPI fine-tune dataset). The results for JOAO and JOAOv2 are taken from [70]. The experimental setting follows [70].

| Dataset | NCI1 | PROTEINS | DD | COLLAB | RDT-B | RDT-M5K |
|---|---|---|---|---|---|---|
| JOAO | 74.48±0.27 | 72.13±0.92 | 75.69±0.67 | 75.30±0.32 | 88.14±0.25 | 52.83±0.54 |
| JOAOv2 | 74.86±0.39 | 73.31±0.48 | 75.81±0.73 | 75.53±0.18 | 88.79±0.65 | 52.71±0.28 |
| AD-GCL-FIX | 75.18±0.31 | 73.96±0.47 | 77.91±0.73 | 75.82±0.26 | 90.10±0.15 | 53.49±0.28 |

Table 12: Semi-supervised Learning with 10% label rate showing 10-Fold Accuracy (%). The results for JOAO and JOAOv2 are taken from [70]. The experimental setting follows [70].

# I Limitations and Broader Impact

We stress on the fact that self-supervised methods come with a fundamental set of limitations as they don't have access to the downstream task information. Specifically for contrastive learning, the design of contrastive pairs (done through augmentations) plays a major role as it guides the encooder to selectively capture certain invariances with the hope that it can be beneficial to downstream tasks. Biases could creep in during the design of such augmentations that can be detrimental to the downstream tasks and risk learning of sub-optimal or non-robust representations of input data. Our work helps to alleviate some of the issues of hand designed augmentation techniques and provides a novel principle that can aid in the design of learnable augmentations. It also motivates further research into the understanding the inherent biases of family of augmentations and how they affect the downstream tasks. Finally, self-supervised graph representation learning has a lot of implications in terms of either fairness, robustness or privacy for the various fields that have been increasing adopting these methods.

# J Compute Resources

All our experiments are performed on a compute cluster managed by Slurm Workload Manager. Each node has access to a mix of multiple Nvidia GeForce GTX 1080 Ti (12GB), GeForce GTX TITAN X (12GB) and TITAN Xp (12GB) GPU cards.