# OpenReview forum: "Adversarial Graph Augmentation to Improve Graph Contrastive Learning"
_NeurIPS.cc/2021/Conference — NeurIPS 2021 Poster_

### Official Review · Reviewer_uwpc · 2021-07-14

**Rating:** 6
**Confidence:** 4

**Summary:**

The paper proposes an adversarial graph augmentation technique to boost graph contrastive learning (GCL), with the purpose of avoiding capturing redundant information to be robust. The proposed technique is driven by the information bottleneck principle (IB) with certain theoretical guarantee and empirical verification.

**Limitations And Societal Impact:**

Not applicable.

**Main Review:**

(Originality)
The paper looks innovative to me.

(Quality)
The overall idea and presentation are clear with extensive experiments on molecule and social network datasets to support rge method.
Theorem 1 is reasonable and informative to show the bound of information the adversarial technique captures. Nevertheless, the derivation is straightforward (combination of the property of infectivity and mutual information rewriting), and therefore it is suggested to downgrade it into proposition to avoid backfire, while it still reveals the key insight of the proposed AD-GCL method.

(Optional clarification)
Would you clarify the contributions in this paper between a latest paper "[Graph Contrastive Learning Automated](https://arxiv.org/abs/2106.07594)" also leveraging adversarial training on GCL?


**Time Spent Reviewing:**

3

---

> ### Author Response · Authors · 2021-08-11
> **Clarification between recent paper "Graph Contrastive Learning Automated" [1] and our AD-GCL.**
>
> Firstly, we thank the reviewer for providing valuable comments and appreciating our work.
>
> The recent paper [1] adopts a min-max optimization framework, which shares some high-level ideas with our adversarial training strategy. However, there are many different aspects. Theoretically, we derive the min-max principle in a more principled way by illustrating its connection to graph information bottleneck (Theorem 1), which explains the fundamental reason and benefits from optimizing graph augmentation strategies. Regarding instantiation, the graph augmentation search spaces of the two works are very different: [1] sets the graph augmentation search space as different types of augmentation that use uniform perturbation, such as uniform edge dropping, uniform node dropping etc. We focus on one type of augmentation (edge dropping) but allow it to be very non-uniform. Actually, our search space is much larger than the search space of [1], in the mathematical sense of uncountable vs. countable spaces. Such a difference also makes the graph augmentation search optimization substantially different. [1] relaxes such a combinatorial  optimization into continuous space by using Jensen’s inequality and solves it via projected gradient descent. Ours, instead, adopts Bayesian modeling plus reparameterization tricks, and gradient descent to optimize. [1]’s technique is good for shallow models where the search space is not complicated, and cannot be applied to search the complicated non-uniform edge-dropping probabilities. That having been said, we can compare the performance between ours and [1] as follows,
>
> ## Unsupervised Learning (SVM downstream classifier and same standard setup as used in [1])
>
> | Dataset    	|    NCI1    	|  PROTEINS  	|     DD     	|    MUTAG   	|   COLLAB   	|    RDT-B   	|   RDT-M5K  	|   IMDB-B   	|
> |------------	|:----------:	|:----------:	|:----------:	|:----------:	|:----------:	|:----------:	|:----------:	|:----------:	|
> | JOAO       	| 78.07±0.47 	| 74.55±0.41 	| 77.32±0.54 	| 87.35±1.02 	| 69.50±0.36 	| 85.29±1.35 	| 55.74±0.63 	| 70.21±3.08 	|
> | JOAOv2     	| 78.36±0.53 	| 74.07±1.10 	| 77.40±1.15 	| 87.67±0.79 	| 69.33±0.34 	| 86.42±1.45 	| 56.03±0.27 	| 70.83±0.25 	|
> | AD-GCL-FIX 	| 75.77±0.50 	| 75.04±0.48 	| 75.38±0.41 	| 88.62±1.27 	| 74.79±0.41 	| 92.06±0.42 	| 56.24±0.39 	| 71.49±0.98 	|
>
> ## Transfer Learning
>  Pre-Training done using ZINC 2M (used for first 8 fine-tune datasets) and PPI-306K (for the last PPI fine-tune dataset)
>
> | Fine-Tune Dataset 	|    BBBP    	|    Tox21   	|    SIDER   	|   ClinTox  	|    BACE    	|     HIV    	|     MUV    	|   ToxCast  	|     PPI    	|
> |-------------------	|:----------:	|:----------:	|:----------:	|:----------:	|:----------:	|:----------:	|:----------:	|:----------:	|:----------:	|
> | JOAO              	| 70.22±0.98 	| 74.98±0.29 	| 59.97±0.79 	| 81.32±2.49 	| 77.34±0.48 	| 76.73±1.23 	| 71.66±1.43 	| 62.94±0.48 	| 64.43±1.38 	|
> | JOAOv2            	| 71.39±0.92 	| 74.27±0.62 	| 60.49±0.74 	| 80.97±1.64 	| 75.49±1.27 	| 77.51±1.17 	| 73.67±1.00 	| 63.16±0.45 	| 63.94±1.59 	|
> | AD-GCL-FIX        	| 70.01±1.07 	| 76.54±0.82 	| 63.28±0.79 	| 79.78±3.52 	| 78.51±0.80 	| 78.28±0.97 	| 72.30±1.61 	| 63.07±0.72 	| 68.83±1.26 	|
>
> ## Semi-supervised Learning (10% label rate)
>
> | Dataset    	|    NCI1    	|  PROTEINS  	|     DD     	|   COLLAB   	|    RDT-B   	|   RDT-M5K  	|
> |------------	|:----------:	|:----------:	|:----------:	|:----------:	|:----------:	|:----------:	|
> | JOAO       	| 74.48±0.27 	| 72.13±0.92 	| 75.69±0.67 	| 75.30±0.32 	| 88.14±0.25 	| 52.83±0.54 	|
> | JOAOv2     	| 74.86±0.39 	| 73.31±0.48 	| 75.81±0.73 	| 75.53±0.18 	| 88.79±0.65 	| 52.71±0.28 	|
> | AD-GCL-FIX 	| 75.18±0.31 	| 73.96±0.47 	| 77.91±0.73 	| 75.82±0.26 	| 90.10±0.15 	| 53.49±0.28 	|
>
> All values for JOAO and JOAOv2 are taken from [1].
>
> [1] You, Yuning, et al. "Graph Contrastive Learning Automated." arXiv preprint arXiv:2106.07594 (2021).

---

### Official Review · Reviewer_PF2i · 2021-07-15

**Rating:** 6
**Confidence:** 4

**Summary:**

In this paper, the authors try to improve the graph contrastive learning. They point out that the representation learned by maximizing mutual information does not necessarily help for downstream tasks. And they propose adversarial graph augmentation to enhance the InfoMax objective for self-supervised learning of GNNs. They argue that AD-GCL will make GNNs avoid learning the redundant information during the training, which are not beneficial to GraphCL. The author also gave a relatively sufficient theoretical explanation. Extensive experiments are conduct to demonstrate the effectiveness of the proposed method.

**Limitations And Societal Impact:**

Please refer to the above questions.

**Main Review:**

In general, the paper is written well and easy to follow. However, there are still several concerns that the authors need to address:

1. Novelty are limited.
(1) Recently, there have been many works [2,3] that have studied the use of high quality data augmentation on graph to improve the performance of contrast learning. The novelty of this work that only improves the performance of graph representation learning by how to drop the edges is very limited.
(2) The method proposed by the author is only to drop the edges, and does not discuss some common graph data augmentation methods, such as dropping nodes, extracting subgraphs and other diverse data augmentation methods mentioned in [1], so I think this data augmentation method is too single, too simple, and very limited in novelty.

2. Questions about the GIB and InfoMax.
(1) The example of failure given in Figure 2 is not credible, because self-supervised learning does not require label information, but it forces the use of random labels to force self-supervised learning to use wrong information while ensuring maximum mutual information. The label information destroys the learned representation. This is unlikely to happen in practical applications, or this is a very extreme situation. I hope that the author can use a more practical example to illustrate that the maximization of mutual information may not necessarily helpful for downstream tasks.
(2) I think the core contribution of this work is what the author mentioned in lines 138-140. It does not need to know the label information of downstream tasks and can capture the minimal information for each graph. And the author's proposed method does not need labels for downstream tasks, but the left and right sides of the equation in Theorem 1 both contain label information. How did the author optimize to remove the label influence.
(3) Will the adversarial data augmentation method proposed by the author have an impact on the maximization of mutual information? In other words, it is impossible to conduct contrastive learning better because of the adversarial examples. I hope the author can give a detial comparison of the accuracy of contrastive learning.

3. Questions about the data augmentation.
(1) The framework proposed by the author only uses a learning method to drop edges, but it cannot perform data augmentation similar to the other methods mentioned in [1].
Simply choosing which edges need to be dropped is definitely not the best choice for graph data.
For graph classification tasks, there are many nodes, node features and even some subgraphs can also produce adversarial data augmentation, but the framework only cover on a small part of the search space, which undoubtedly ignored a lot of other possibilities.
(2) As mentioned in [1], dropping edges is not effective for all datasets. For example, for some molecular data sets such as biochemistry, the connection relationship is a very important property for this kind of data. If you let GNN learn this invariance, it will have a negative impact on downstream tasks. Does the author also find this same rule in the experiment? If found, please give detailed experimental results, if not found, please explain the reasons.

4. Questions about the experiments.
(1) The paper is missing a lot of baselines[2,3], please compare your experimental results with these methods, and point out the superiority of your method and some of the shortcomings of their methods.
(2) The paper only conducted experiments on many small data sets. May I ask whether it is also effective for large data sets such as OGB datasets [4]. If so, please give more experimental results on large datasets.
(3) The paper just performed a lot of experiments on graph classification tasks. May I ask whether it is also effective for node classification or link prediction tasks. If it is effective, please provide some experimental results. If it is not effective, please give an explanation.

[1] Graph contrastive learning with augmentations

[2] Graph Contrastive Learning with Adaptive Augmentation

[3] Graph Contrastive Learning Automated

[4] https://ogb.stanford.edu/

**Time Spent Reviewing:**

5h

---

> ### Author Response · Authors · 2021-08-11
> **Response to Reviewer 3 (Part 1)**
>
> ## Q 1.2, 3.1 Limited novelty & Limited types of data augmentation
>
> Reviewer 3 seems to assume that building a graph contrastive learning model should always extensively study different ways or combinations of graph augmentation methods before claiming a good learning strategy. Because of this assumption, Reviewer 3 thinks our study that uses edge dropping is of limited novelty and provides an ineffective graph contrastive learning model.
>
> We respectfully disagree with the above unwarranted assumption and the wrong conclusion. First, we greatly appreciate previous works such as [1, 2] which stand from an angle of empirical study and do compare different ways of graph augmentations.  However, for a real application, especially those with few labels that favor graph contrastive learning, labels are insufficient for a developer to design and test different types of graph augmentation. No one guarantees that the type of graph augmentation that works for the public datasets also works for real application datasets. Our work is motivated by this observation and decides to propose a fundamental principle that may generally benefit graph contrastive learning without having to cherry pick the type of graph augmentation. To show the proof of concept, we focus on one dataset-independent augmentation, edge-dropping, to demonstrate the general effectiveness of AD-GCL on multiple datasets.
>
> Our results successfully demonstrate that it is possible to use one type of graph augmentation to achieve very good performance, and the benefit uniquely comes from being optimized under the AD-GCL principle. We show that even for a very “single”, “simple”, “limited” (copied from the words of Reviewer 3) type of graph augmentation, edge dropping, AD-GCL makes it outperform previous works that leverage extensive/different combinations of graph augmentations. This result actually demonstrates the significance of AD-GCL not the other way around. We believe that if different types of graph augmentation are allowed, AD-GCL may only yield better performance. In the end, we want to say a general argument that the novelty of a research work is never about how various scenarios it has discussed but more about what problem it can indeed solve.
>
> ## Q 3.2 Edge dropping as the graph augmentation method could be problematic
>
> Reviewer 3 refers to [1] and seems to make an incorrect assumption that the conclusion obtained by adopting uniform edge dropping is still applied to our case with non-uniform edge dropping where the non-uniform probability is given by adversarial training.
>
> First, our experimental results directly indicate that Reviewer 3 is wrong. Non-uniform edge dropping trained via AD-GCL actually outperforms GraphCL [1] that uses the manually-chosen and dataset specific data augmentation on 14 (classification + regression tasks) benchmark datasets for unsupervised learning. To clearly demonstrate the difference between uniform edge dropping and non-uniform edge dropping, we perform the very important ablation NAD-GCL (non-adversarial) that performs uniform edge dropping graph augmentation. Clearly, non-uniform edge dropping assists the model to achieve much better performance than the uniform-edge-dropping counterpart in all benchmarks datasets (Table 1) and Figure 3c even looks at edge dropping of different dropping probabilities for an even granular analysis.
>
> Then, a natural question is what fundamentally makes the learnable edge dropping different. [1] argues that uniform edge dropping may hurt the performance on molecule datasets because molecule graphs have important structural fingerprints that may get lost in edge dropping. However, this argument does not apply to non-uniform edge dropping because non-uniform edge dropping may actually emphasize such structural fingerprints by dropping more noisy structures.
>
> [1] You, Yuning, et al. "Graph contrastive learning with augmentations." NeurIPS 2020
>
> [2] Zhu, Yanqiao, et al. "Graph contrastive learning with adaptive augmentation." Proceedings of the Web Conference 2021. 2021.
>
> [3] You, Yuning, et al. "Graph Contrastive Learning Automated." arXiv preprint arXiv:2106.07594 (2021).
>
> [4] Wu, Felix, et al. "Simplifying graph convolutional networks." International conference on machine learning. PMLR, 2019.
>
> [5] Klicpera, Johannes, Aleksandar Bojchevski, and Stephan Günnemann. "Predict then propagate: Graph neural networks meet personalized pagerank." arXiv preprint arXiv:1810.05997 (2018).
>
> [6] Chien, Eli, et al. "Adaptive universal generalized pagerank graph neural network." arXiv preprint arXiv:2006.07988 (2020).
>
> [7] Xu, Keyulu, et al. "How powerful are graph neural networks?." arXiv preprint arXiv:1810.00826 (2018).
>
> [8] Zhang, Muhan, et al. "Revisiting graph neural networks for link prediction." arXiv preprint arXiv:2010.16103 (2020).
>
> [9] Zhang, Muhan, and Yixin Chen. "Link prediction based on graph neural networks." Advances in Neural Information Processing Systems 31 (2018): 5165-5175.
>
> [10] Sun, Fan-Yun, et al. "Infograph: Unsupervised and semi-supervised graph-level representation learning via mutual information maximization." ICLR 2020
>
> [11] Tian, Yonglong, et al. "What makes for good views for contrastive learning?." NeurIPS 2020
>
> [12] Tschannen, Michael, et al. "On mutual information maximization for representation learning." ICLR 2020.

---

> ### Author Response · Authors · 2021-08-11
> **Response to Reviewer 3 (Part 2)**
>
> ## Q 1.1, 4.1 Insufficient baselines
>
> Reviewer 3 doubts our novelty as opposed to the baselines [2][3] and asks us to further compare with [2][3].
>
> We compare AD-GCL with [2]. Regarding the problem setting, [2] was proposed only for performing unsupervised node classification, while our method and experiments are for unsupervised, transfer and semi-supervised graph classification and regression. Second, regarding the methodology, [2] leverages node centrality measures to define edge dropping augmentations. The reason why they choose node centrality measures is because their node classification task is performed on highly homophilic networks, where node centrality has a great impact on the homophilic sections of the networks. From this perspective, so-called adaptive augmentation strategies in [2] are actually strongly coupled with their applications i.e., node classification. That strategy is hard to apply to chemical and biological molecule graphs for graph-level tasks. In contrast, our adaptive augmentation strategy is designed in a very different way, which comes from more fundamental adversarial-training dynamics that is independent of the applications.
>
> We compare AD-GCL with [3]. First, note that this paper was released online (arxiv, Jun 28, 2021) one month after the NeurIPS submission deadline (May 28, 2021) so it does not make sense to let us cite and discuss [3] in the under-review version, though we are happy to discuss the difference in the camera-ready version. At a first glance, [3] adopts an min-max optimization framework, which shares some high-level ideas with our adversarial training strategy. However, there are many different aspects. Theoretically, we derive the min-max principle in a more principled way by illustrating its connection to graph information bottleneck (Theorem 1), which explains the fundamental reason and benefits from optimizing graph augmentation strategies. Second, regarding instantiation, the graph augmentation search spaces of the two works are very different: [3] sets the graph augmentation search space as different types of augmentation that use uniform perturbation, such as uniform edge dropping, uniform node dropping etc. We focus on one type of augmentation (edge dropping) but allow it to be very non-uniform. Actually, our search space is much larger than the search space of [3], in the mathematical sense of uncountable vs.. countable, so Reviewer 3 is again wrong in this sense. Such a difference also makes the graph augmentation search optimization substantially different. [3] relaxes such a combinatorial  optimization into continuous space by using Jensen’s inequality and projected gradient descent. Ours, instead, adopts Bayesian modeling plus reparameterization tricks, and gradient descent to optimize. [3]’s technique is good for shallow models where the search space is not complicated, and cannot be applied to search the complicated non-uniform edge-dropping probabilities like our case. That having been said, we can compare the performance between ours and [3] as follows, though [3] is online after we submitted our work.
>
> ### Unsupervised Learning (SVM downstream classifier and same standard setup as used in [1])
>
> | Dataset    	|    NCI1    	|  PROTEINS  	|     DD     	|    MUTAG   	|   COLLAB   	|    RDT-B   	|   RDT-M5K  	|   IMDB-B   	|
> |------------	|:----------:	|:----------:	|:----------:	|:----------:	|:----------:	|:----------:	|:----------:	|:----------:	|
> | JOAO       	| 78.07±0.47 	| 74.55±0.41 	| 77.32±0.54 	| 87.35±1.02 	| 69.50±0.36 	| 85.29±1.35 	| 55.74±0.63 	| 70.21±3.08 	|
> | JOAOv2     	| 78.36±0.53 	| 74.07±1.10 	| 77.40±1.15 	| 87.67±0.79 	| 69.33±0.34 	| 86.42±1.45 	| 56.03±0.27 	| 70.83±0.25 	|
> | AD-GCL-FIX 	| 75.77±0.50 	| 75.04±0.48 	| 75.38±0.41 	| 88.62±1.27 	| 74.79±0.41 	| 92.06±0.42 	| 56.24±0.39 	| 71.49±0.98 	|
>
> ### Transfer Learning
>  Pre-Training done using ZINC 2M (used for first 8 fine-tune datasets) and PPI-306K (for the last PPI fine-tune dataset)
>
> | Fine-Tune Dataset 	|    BBBP    	|    Tox21   	|    SIDER   	|   ClinTox  	|    BACE    	|     HIV    	|     MUV    	|   ToxCast  	|     PPI    	|
> |-------------------	|:----------:	|:----------:	|:----------:	|:----------:	|:----------:	|:----------:	|:----------:	|:----------:	|:----------:	|
> | JOAO              	| 70.22±0.98 	| 74.98±0.29 	| 59.97±0.79 	| 81.32±2.49 	| 77.34±0.48 	| 76.73±1.23 	| 71.66±1.43 	| 62.94±0.48 	| 64.43±1.38 	|
> | JOAOv2            	| 71.39±0.92 	| 74.27±0.62 	| 60.49±0.74 	| 80.97±1.64 	| 75.49±1.27 	| 77.51±1.17 	| 73.67±1.00 	| 63.16±0.45 	| 63.94±1.59 	|
> | AD-GCL-FIX        	| 70.01±1.07 	| 76.54±0.82 	| 63.28±0.79 	| 79.78±3.52 	| 78.51±0.80 	| 78.28±0.97 	| 72.30±1.61 	| 63.07±0.72 	| 68.83±1.26 	|
>
> ### Semi-supervised Learning (10% label rate)
>
> | Dataset    	|    NCI1    	|  PROTEINS  	|     DD     	|   COLLAB   	|    RDT-B   	|   RDT-M5K  	|
> |------------	|:----------:	|:----------:	|:----------:	|:----------:	|:----------:	|:----------:	|
> | JOAO       	| 74.48±0.27 	| 72.13±0.92 	| 75.69±0.67 	| 75.30±0.32 	| 88.14±0.25 	| 52.83±0.54 	|
> | JOAOv2     	| 74.86±0.39 	| 73.31±0.48 	| 75.81±0.73 	| 75.53±0.18 	| 88.79±0.65 	| 52.71±0.28 	|
> | AD-GCL-FIX 	| 75.18±0.31 	| 73.96±0.47 	| 77.91±0.73 	| 75.82±0.26 	| 90.10±0.15 	| 53.49±0.28 	|
>
> All values for JOAO and JOAOv2 are taken from [1].
>
>
>
> ## Q 4.2, 4.3 Limited experiments
>
> Reviewer 3 asks us to perform experiments on OGB datasets and also perform node classification/link prediction experiments.
>
> Reviewer 3 points out that our evaluation doesn’t include large OGB [4] datasets. On the contrary we do include large OGB datasets for unsupervised (Table 1) and transfer learning (Table 2), while previous works [1,2] missed perhaps because of the scalability constraints. There is a computation bottleneck for evaluating unsupervised graph representations on these OGB datasets, which lies in the final prediction evaluation protocol. It is standard practice to train a separate predictor (e.g., non-linear svm classifier by the LBFGS solver)  that leverages the unsupervised representations to predict graph labels. It usually takes a very long time for these large OGB datasets especially when the representations have a large size (e.g. 300 dim size). We highlight that the community needs alternate evaluation protocols that can scale to evaluating graph representation learning techniques to large dataset sizes. For these large and high-dimensional OGB datasets, we adopt a linear predictor to guarantee the scalability.
>
> Regarding extending our adversarial training framework to node classification and link prediction tasks, we are happy to do so in the future work but we feel it is too much and distracting to merge it into the current paper. The current paper of 9 pages has already been very crowded.
>
> Moreover, we would like to argue against the tendency of the GNN community to ask for a unique GNN work to solve all graph learning tasks. Many recent works have shown that node classification, link prediction and graph classification essentially depend on substantially different properties of the graphs. For example, node classification typically depends on homophilic/heterophilic  node features and community structures (SGC [4], APPNP [5] and GPRGNN [6]); Link prediction depends on local dense subgraphs connecting two nodes [SEAL[9], 8]; graph classification depends on distinguishable representations of non-isomorphic graphs  (GIN [7]). These differences require GNNs to emphasize different aspects to achieve the optimal performance, where evaluating a single model on tasks that need fundamentally different properties may simply flood the key benefit. For example, GIN is good for graph classification due to the expressive power but is not good for node classification, GCN is good for node classification due to node feature denoising but is not good for graph classification [7], GIN and GCN are both not good for link prediction as they miss capturing the joint structure between two nodes [8, 9]. Given this observation, we do not want to merge all the tasks together, while we believe our adversarial training framework can be generalized to those node level tasks, though more investigation on the proper backbone encoders and implementations is needed.
>
> [1] You, Yuning, et al. "Graph contrastive learning with augmentations." NeurIPS 2020
>
> [2] Zhu, Yanqiao, et al. "Graph contrastive learning with adaptive augmentation." Proceedings of the Web Conference 2021. 2021.
>
> [3] You, Yuning, et al. "Graph Contrastive Learning Automated." arXiv preprint arXiv:2106.07594 (2021).
>
> [4] Wu, Felix, et al. "Simplifying graph convolutional networks." International conference on machine learning. PMLR, 2019.
>
> [5] Klicpera, Johannes, Aleksandar Bojchevski, and Stephan Günnemann. "Predict then propagate: Graph neural networks meet personalized pagerank." arXiv preprint arXiv:1810.05997 (2018).
>
> [6] Chien, Eli, et al. "Adaptive universal generalized pagerank graph neural network." arXiv preprint arXiv:2006.07988 (2020).
>
> [7] Xu, Keyulu, et al. "How powerful are graph neural networks?." arXiv preprint arXiv:1810.00826 (2018).
>
> [8] Zhang, Muhan, et al. "Revisiting graph neural networks for link prediction." arXiv preprint arXiv:2010.16103 (2020).
>
> [9] Zhang, Muhan, and Yixin Chen. "Link prediction based on graph neural networks." Advances in Neural Information Processing Systems 31 (2018): 5165-5175.
>
> [10] Sun, Fan-Yun, et al. "Infograph: Unsupervised and semi-supervised graph-level representation learning via mutual information maximization." ICLR 2020
>
> [11] Tian, Yonglong, et al. "What makes for good views for contrastive learning?." NeurIPS 2020
>
> [12] Tschannen, Michael, et al. "On mutual information maximization for representation learning." ICLR 2020.

---

> ### Author Response · Authors · 2021-08-11
> **Response to Reviewer 3 (Part 3)**
>
> ## Q 2.1, 2.2, 2.3 Confusions on the theory, regarding GIB and InfoMax
>
> Reviewer 3 gets confused about the motivating example in Sec. 3.1, the implication of Theorem 1 and the benefit that adversarial training brings to InfoMax. We clarify them as follows.
>
> Reviewer 3 argues that our motivational experiment in Sec 3.1 figure 2 is not credible because performing information maximization while simultaneously providing random label supervision doesn’t occur in practice and “destroys” learned representations. However, the goal of the experiment is to show that the “destroyed” learned representations can still maximize mutual information and identify each graph in the dataset, which indicates that only using mutual information maximization could be problematic. We show that with a mutual information maximization objective and random label supervision, the representation quality is of no use for the true label prediction downstream task (so the encoder has poor representation quality) while it can still identify each graph in the dataset (that means the encoder maximizes the mutual information of the input dataset). This clearly shows that the InfoMax principle may not be a reliable way to learn representations. Moreover, multiple other papers (InfoMin [11] and [12]) have also shown a similar phenomenon in the image domain, where we demonstrate it in the graph learning domain. Complete details of the motivations and experiments are provided in the Appendix G.1 of our paper.
>
> Reviewer 3 also incorrectly interprets Theorem 1. We clarify that, Theorem 1 doesn’t show how to perform the optimization, but Eq. 6 does. The sole reason for Theorem 1 is to show how the graph encoder learned by using AD-GCL (Eq.6) behaves w.r.t. task irrelevant information and task relevant information. We make use of labels Y only to quantify such information. Then, in practice, how does the model avoid using the information of label Y? It is all characterized by the graph augmentation search space $\mathcal{T}$. The first statement of Theorem 1 indicates AD-GCL reasonably approximates GIB by effectively removing irrelevant information. The second statement of Theorem 1 indicates that the graph augmentation search space $\mathcal{T}$ has to be properly controlled (later implemented by regularization later) to avoid losing too much information related to the downstream tasks.
>
> Misinterpreting the motivational experiments and Theorem 1 further pushes Reviewer 3 to question about the effect of adversarial training on mutual information maximization. Reviewer 3’s observation that adversarial training may make it hard to maximize mutual information is correct, but the assumption made by Reviewer 3 that mutual information should be maximized is wrong. The motivational experiment in Sec 3.1 Fig. 2 and the GIB principle are to tell that the encoder should not do mutual information maximization. We argue that the encoder should capture minimal sufficient information but not maximal mutual information. Our extensive experiments just demonstrate that AD-GCL that favors the minimal sufficient information will significantly outperform GraphCL [1] and InfoGraph [10] baselines that favor mutual information maximization.
>
>
> [1] You, Yuning, et al. "Graph contrastive learning with augmentations." NeurIPS 2020
>
> [2] Zhu, Yanqiao, et al. "Graph contrastive learning with adaptive augmentation." Proceedings of the Web Conference 2021. 2021.
>
> [3] You, Yuning, et al. "Graph Contrastive Learning Automated." arXiv preprint arXiv:2106.07594 (2021).
>
> [4] Wu, Felix, et al. "Simplifying graph convolutional networks." International conference on machine learning. PMLR, 2019.
>
> [5] Klicpera, Johannes, Aleksandar Bojchevski, and Stephan Günnemann. "Predict then propagate: Graph neural networks meet personalized pagerank." arXiv preprint arXiv:1810.05997 (2018).
>
> [6] Chien, Eli, et al. "Adaptive universal generalized pagerank graph neural network." arXiv preprint arXiv:2006.07988 (2020).
>
> [7] Xu, Keyulu, et al. "How powerful are graph neural networks?." arXiv preprint arXiv:1810.00826 (2018).
>
> [8] Zhang, Muhan, et al. "Revisiting graph neural networks for link prediction." arXiv preprint arXiv:2010.16103 (2020).
>
> [9] Zhang, Muhan, and Yixin Chen. "Link prediction based on graph neural networks." Advances in Neural Information Processing Systems 31 (2018): 5165-5175.
>
> [10] Sun, Fan-Yun, et al. "Infograph: Unsupervised and semi-supervised graph-level representation learning via mutual information maximization." ICLR 2020
>
> [11] Tian, Yonglong, et al. "What makes for good views for contrastive learning?." NeurIPS 2020
>
> [12] Tschannen, Michael, et al. "On mutual information maximization for representation learning." ICLR 2020.

---

### Official Review · Reviewer_dRac · 2021-07-16

**Rating:** 6
**Confidence:** 4

**Summary:**

Current contrastive learning (CL) methods are based on various graph augmentations to generate multiple views for contrast (maximizing MI). To discover the optimal augmentations, the authors propose an adversarial approach to automatically learn views generation that tries to minimize the MI between two views.

**Limitations And Societal Impact:**

The authors have discussed the limitations and societal impact in the Appendix.

**Main Review:**

This paper proposed a novel method leveraging adversarial training to learn augmentations for graph contrastive learning. They also introduce edge-dropping augmentation for graph-level tasks as an example to show the theoretical base of the model and the outperforming of their model compared with other popular graph contrastive methods.

The proposed method is overall novel and interesting. However, there are major concerns in the experimental results. In Table 2, 8, 9 10, the performance gain of the proposed method is not consistent among datasets, especially when comparing to its direct baseline (GraphCL). This makes the author's claim and the proposed method less convincing. The authors may need more experiments to show the effectiveness of their methods on different tasks.

Below are additional questions in need of clearance.

1, In Section 3.2 (Learnable Edge Dropping GDA model T_\Phi()), the authors firstly claimed that “structure fingerprints” are important for downstream tasks and then claimed that dropping some edges will not change subgraph structure. The two claims seem conflicting with each other as one type of subgraph can be turned into another type of subgraph by dropping some edges, then the contrastive learning method cannot distinguish these two types of subgraphs. This situation is common in the chemical analysis of molecules, so the authors may need to introduce additional assumptions or proof of this question if they want to mainly illustrate their method by edge-dropping augmentation.

2, In Section 3.2 (Parameterizing T_\Phi()), the authors called their T as a “graph model”, this was confusing because it was not consistent with the terminology used in Section 3.1, especially in Definition 1 and Definition 2. Do the authors mean “instance of graph augmentation model”?

3, In Section 3.2 (Regularizing T_\Phi()), the authors proposed to add a regularization term to control the edge perturbation aggressiveness. This is a good idea and classic method in machine learning, but I have two detailed questions here. First, even with the regularization term, it can still have some augmented graphs without any edges in training. Is this situation reasonable or how do the authors prevent it? Second, adding this regularization term was for “not to perform very aggressive perturbation”, but in the Experiment details (Appendix F), the authors showed that aggressive perturbation was somehow better. Let us consider a situation where only the node feature suffices IB or InfoMin target, then the above situation (3.1) is reasonable and can be common sense. The author needs to make it more convincing the necessity of using the regularization term, the edge dropping method, or even GNNs in these experiments.

4, Although the authors showed that even only learnable edge dropping augmentation could lead to SOTA results, the Claims and Proofs in Section 3.1 were not limited in edge dropping and it seems not very easy to analogy edge drooping to other augmentation methods like node dropping and feature masking. The authors maybe need some brief explanation of why this method is not limited to edge-dropping augmentation and suitable tasks for edge-dropping. Also, it would be better to include more experiments to show their learnable augmentation method can be easily transferred to other augmentation methods and suitable tasks for them.

5, Typo at line 137



===================

I'd like to update my score after reading the authors' responses and comments from other reviewers.

**Time Spent Reviewing:**

3.5

---

> ### Author Response · Authors · 2021-08-11
> **Response to Reviewer 2**
>
> Firstly, we thank the reviewer for providing feedback and constructive comments. Several interesting questions related to our augmentation and regularization were raised and we provide detailed answers to each one of them below.
>
> ## Q1. Edge dropping as the graph augmentation method could be problematic
>
> Non-uniform edge dropping trained via AD-GCL actually outperforms GraphCL [1] that uses the manually-chosen and dataset specific data augmentation on 14 (classification + regression tasks) benchmark datasets for unsupervised learning. To clearly demonstrate the difference between uniform edge dropping and non-uniform edge dropping, we perform the very important ablation NAD-GCL (non-adversarial) that performs uniform edge dropping graph augmentation. Clearly, non-uniform edge dropping assists the model to achieve much better performance than the uniform-edge-dropping counterpart in all benchmarks datasets (Table 1) and Figure 3c even looks at edge dropping of different dropping probabilities for an even granular analysis.
>
> Then, a natural question is what fundamentally makes the learnable edge dropping different. [1] argues that uniform edge dropping may hurt the performance on molecule datasets because molecule graphs have important structural fingerprints that may get lost in edge dropping. However, this argument does not apply to non-uniform edge dropping because non-uniform edge dropping may actually emphasize such structural fingerprints by dropping more noisy structures.
>
>
> ## Q2. Clarifying the definition of $T_\Phi$ In Section 3.2  Do the authors mean “instance of graph augmentation model”?
>
> Yes, it can be seen like that. The $T_\Phi$ in section 3.2 takes the role of a learnable edge-dropping model. It is to be seen as a member of $\mathcal{T}$ which is the set of all possible GDA families that can be defined.
>
> ## Q3. Even with the regularization term, it can still have some augmented graphs without any edges in training. How is this reasonable ?
>
> We would like to respond by first making it clear that in addition to the regularization term, there is a hyper-parameter called $\lambda_{reg}$ that controls the strength of the regularization term. We provide detailed sensitivity analysis of $\lambda_{reg}$ w.r.t expected number of edges dropped in Figure 4 in Appendix. From those plots, it is clear that higher values of $\lambda_{reg}$ (say 5 +) will surely keep > 70-80% of the edges in the augmented graph. The situation the reviewer mentions will arise only when $\lambda_{reg}$ is extremely small which is not recommended. This also motivates the need for regularization. As $\lambda_{reg}$ value of 0 (i.e. no regularization), results in almost all edges being dropped. Our ablation of AD-GCL-FIX vs AD-GCL-OPT looks at the representation quality of our principle with a fixed large lambda of 5 (FIX) and one that is optimized on the validation set (OPT). Interestingly, all the OPT $\lambda_{reg}$ values tend to be large (closed to or larger than 5). This is clearly shown in Table 7.
>
> There are some exceptions. Interestingly, for example in regression datasets like ogbg-mollipo, OPT $\lambda_{reg}$ value is quite low 0.1, but it doesn't result in all edges being dropped (> 40% of edges are retained. See Figure 4, ogbg-mollipo plot. (1st row, 2nd column)). This phenomenon is witnessed particularly in regression tasks. We would like to further investigate this phenomenon in the future.
>
> We also look for more answers and provide the analysis of how the training dynamics affect the expected number of edges dropped (Appendix, Figure 7 ). It shows that high $\lambda_{reg}$ values are more stable than lower ones, moreover, even lower $\lambda_{reg}$ values that initially result in large edge drops finally converge to reasonably low edge drop ratios. Suggesting that the training dynamics play a vital role in the AD-GCL paradigm. Although it gives an interesting picture, more theoretical studies are required to fully understand this phenomenon.
>
> ## Q4. Limited applicability of AD-GCL to other graph augmentations
>
> Reviewer 2 questions about the applicability of AD-GCL on other types of augmentations.
>
> Empirically, the techniques that apply AD-GCL to edge dropping can be easily extended to  other types of augmentations, by using Bayesian deep learning approaches and reparameterization methods. Taking node dropping as an example, we still use GNN as the augmentor to learn node representations, then use these representations to generate reparameterized Bernoulli random variables that are used to drop nodes. The only difference is that edge dropping uses reparameterized Bernoulli random variables based on a pair of nodes while node dropping uses reparameterized Bernoulli random variables based on a single node.
>
> By following the above sense, AD-GCL can be also used to node feature perturbation (by imposing GNN-reparameterized Gaussian random variables on node features), node feature masking (by imposing GNN-reparameterized Bernoulli random variables on node features), and subgraph extraction (by combining the general gumbel-softmax reparameterization tricks [2] and GNNs).
>
> All of these graph augmentations can be merged into the AD-GCL framework, though they may have different computation-accuracy tradeoffs.
>
> [1] You, Yuning, et al. "Graph contrastive learning with augmentations." NeurIPS  (2020)
>
> [2] Paulus, Max B., et al. "Gradient estimation with stochastic softmax tricks."NeurIPS (2020).

---

### Official Review · Reviewer_GWqt · 2021-07-17

**Rating:** 7
**Confidence:** 5

**Summary:**

The paper combines the ideas of adversarial training (on augmentation) and graph contrastive learning (branded as "AD-GCL") to further boost the unsupervised performance, targeting at representing less redundant information which is on pair with information bottleneck principle. It is empirically demonstrated with significant performance gains in multiple experiment settings of unsupervised, semi-supervised and transfer learning.

**Limitations And Societal Impact:**

Nothing in particular

**Main Review:**

Generally good presentation and solid experiments. I like it in general, with a few comments and suggestions:

(i) (Method): Using learnable edge perturbation to parameterize augmentation is a practical way but might be restricted. Can authors provide some other instantiations to consolidate the proposed idea (e.g. learnable node masking)?

(ii) (Theoretical claim): The claim in Theorem 1 holds the assumption seemingly conflicting with the goal of unsupervised learning, that assuming the GNN is already endowed with good enough distinguishment power, while this is difficult to achieve specifically for certain downstream labels since unsupervised learning does not have labels. More clarification of this claim is needed.

(iii) (Experiment): Table 1&3 show AD-GCL achieves the (nearly) best performance among all datasets (including molecules) but Table 2 shows only 3 out of 8 doing better. Is there any explanation (analysis) for these contradictory observations?

**Time Spent Reviewing:**

1

---

> ### Author Response · Authors · 2021-08-11
> **Response to Reviewer 1**
>
> Firstly, we thank the reviewer for providing detailed feedback and constructive comments. Below we respond to all the questions raised by the reviewer.
>
> ## Q1. Applicability of AD-GCL to other graph augmentations
>
> Reviewer 1 questions about the applicability of AD-GCL on other types of augmentations as they think edge dropping might be restrictive.
>
> Empirically, the techniques that apply AD-GCL to edge dropping can be easily extended to  other types of augmentations, by using Bayesian deep learning approaches and reparameterization methods. Taking node dropping as an example, we still use GNN as the augmentor to learn node representations, then use these representations to generate reparameterized Bernoulli random variables that are used to drop nodes. The only difference is that edge dropping uses reparameterized Bernoulli random variables based on a pair of nodes while node dropping uses reparameterized Bernoulli random variables based on a single node.
>
> By following the above sense, AD-GCL can be also used to node feature perturbation (by imposing GNN-reparameterized Gaussian random variables on node features), node feature masking (by imposing GNN-reparameterized Bernoulli random variables on node features), and subgraph extraction (by combining the general gumbel-softmax reparameterization tricks [3] and GNNs).
>
> All of these graph augmentations can be merged into the AD-GCL framework, though they may have different computation-accuracy tradeoffs.
>
>
> ## Q2. Assumption in our theorem 1 that the GNN encoder needs to be as powerful as the 1-WL conflicts with unsupervised representation learning.
>
> Because theorem 1 assumes that the encoder GNN is as powerful as the 1-WL test, the reviewer questions if that conflicts with the goal of unsupervised graph representation learning. The reviewer reasons that since the encoder is already endowed with good distinguishing power to start with, contrastive learning would not be beneficial.
>
> The reviewer provides a very insightful observation. We interpret it as follows: When an encoder can distinguish the examples in the dataset very well, then mutual information has already been maximized and thus no contrastive learning is needed. This is partially correct but saying our assumption thus becomes void is wrong.
>
> First, our assumption that the GNN encoder is as powerful as the 1-WL test does not mean that before training the GNN encoder has already been that powerful. A more precise assumption is that the GNN encoder could be as powerful as the 1-WL test with proper parameters, although most GNN works do not use this way to claim the assumption. This is a common assumption that almost all GNN encoders satisfy, which also includes GIN [1] that our experiments use.
>
> Second, just as claimed as the main motivation of our work, even if mutual information gets maximized, contrastive learning is still needed. Actually, random initialized GIN (one of our baselines termed RU-GIN) most likely has already been as powerful as the 1-WL test on distinguishing graphs in a real dataset with countable features [1] because the number of real graphs is limited. But clearly, RU-GIN’s performance is still much worse than all contrastive learning methods. The reason is because of the geometry of graph representations. Although the representations of the graphs that are distinguishable based on the 1-WL test are distinctive, the geometry of those representations does not well measure the similarity of graphs and thus gives bad performance in the downstream task.  Contrastive learning actually helps tune such geometry. Traditional contrastive learning implements such tuning by using the loss like $I_{NCE}$ [2]. Our paper further claims that this is insufficient. Besides such a loss, we also need to consider information-bottleneck-induced loss, which further tunes the geometry by imposing constraints on the mutual information.
>
>
> ## Q3. Performance gains of AD-GCL in Table 2 are lower compared to Table 1 and 3.
>
> We reason that our gain for transfer learning decreases compared to unsupervised learning, because transfer learning protocol utilizes label information of the fine-tune dataset to fine-tune the encoder parameters after pre-training. Whereas, for Table 1 which shows unsupervised learning setting, no fine-tuning is done in its evaluation protocol and only a linear classifier is trained with the representations given by a fixed encoder. Our intuition suggests that more label information means less dependence on (robust) representations and in turn less dependence on AD-GCL. This reasoning also applies to semi-supervised learning (Table 3) where only 10% of labels are utilized as opposed to 100% label information in transfer learning and 0% in unsupervised learning to tune parameters of the encoder. Thus, we obtain up-to 6% gains for transfer learning compared to 14% in unsupervised settings.
>
>
> [1] Xu, Keyulu, et al. "How powerful are graph neural networks?. "ICLR 2019
>
> [2] Tschannen, Michael, et al. "On mutual information maximization for representation learning. " ICLR 2020.
>
> [3] Paulus, Max B., et al. "Gradient estimation with stochastic softmax tricks."NeurIPS (2020).

---

### Author Response · Authors · 2021-08-11
**Rebuttal Summary**

We thank the reviewers for reading our paper and providing valuable feedback that helps us improve the manuscript. All reviewers agree that our paper is written well and recognize that the problems we are solving are indeed important. We respond to each reviewer separately and carefully, and expect to have more feedback.

---

### Author Response · Authors · 2021-08-27
**Further comparison summary of our AD-GCL and recent work JOAO [1].**

Firstly, we thank again all the reviewers for providing us valuable feedback, comments and reading our paper. Reviewers 3 [PF2i] and 4 [uwpc] asked us to compare our AD-GCL with JOAO [1] in their review comments/questions. Note that JOAO was released after the NeurIPS submission deadline so we did not compare with JOAO in the submission.

In the response to Reviewers 3 [PF2i] and 4 [uwpc] , we have provided comparisons between AD-GCL with JOAO on transfer learning, semi-supervised learning, and unsupervised learning with a kernel SVM as the downstream classifiers (We used JOAO's performance provided by [1]). As we argued that using a complicated downstream classifier may be unfair to compare the quality of the trained GNN encoders. Here we further provide the comparison on unsupervised learning with a linear downstream classifier. The discussion that compares the linear classifier and the non-linear classifier is in the lines 817 – 839 of the appendix of our paper.

Let us first clarify the different mechanisms that JOAO and AD-GCL adopt. JOAO [1] selects augmentation families from a pool $\mathcal{A}$ = {NodeDrop, Subgraph,EdgePert, AttrMask,Identical} and defines a uniform prior on them for their inner optimization over all possible augmentation family pairs. (See Section 3.2 and See Eq. 7,8 in [1]). An important distinction is that JOAO still adopts uniformly random augmentations and the inner optimization only searches over different pairs of uniform augmentations. Whereas, AD-GCL adopts non-uniformly random augmentations, which essentially corresponds to a much larger search space.

Complexity wise, JOAO is more expensive than AD-GCL as, they utilize projected gradient descent to fully optimize the inner optimization step over all possible augmentations $\mathcal{A}$. This is a factor k more expensive than AD-GCL. The factor k in JOAO is currently $|\mathcal{A}|^2 = 4^2 = 16$. This makes it slow to train while still having a restricted search space compared to AD-GCL which on the other hand is both faster and looks at a larger search space for a given augmentation family. In our experiments on a single GPU, JOAO took ~3.2 hrs for training on COLLAB whereas AD-GCL only took ~14.4 mins (0.24 hrs).

Now, we provide the comparison between JOAO and AD-GCL using a linear evaluation protocol for unsupervised setting. Specifically, a linear SVM head is used for evaluating the representations learned by the 2 methods for the downstream task. The regularization hyper-parameter of the linear svm is grid-searched among {0.001, 0.01,0.1,1,10,100,1000}. We re-run the code provided by authors of JOAO (available at [ https://github.com/Shen-Lab/GraphCL_Automated ]) with the default parameters for 5 times each with different seeds. The only change done is to the embedding evaluation code to include linear svm as the final prediction head. For all the TU datasets used here, standard 10-Fold evaluation is used to report classification accuracy (%).

### Unsupervised learning performance using a linear evaluation protocol

| Dataset                 |    NCI1    |  PROTEINS  |     DD     |    MUTAG   |   COLLAB   |    RDT-B   |   RDT-M5K  |   IMDB-B   |
|-------------------------|:----------:|:----------:|:----------:|:----------:|:----------:|:----------:|:----------:|:----------:|
| JOAOv2  (FIX-gamma=0.1) | 72.99±0.75 | 71.25±0.85 | 66.91±1.75 | 85.20±1.64 | 70.40±2.21 | 78.35±1.38 | 45.57±2.86 | 71.60±0.86 |
| AD-GCL-FIX              | 69.67±0.51 | 73.59±0.65 | 74.49±0.52 | 89.25±1.45 | 73.32±0.61 | 85.52±0.79 | 53.00±0.82 | 71.57±1.01 |

The results in the above table further indicate that AD-GCL performs better than JOAO in 6 of the 8 TU benchmark datasets. The gap in performance is even more clear compared to the non-linear evaluation setting as shown previously in the response to Reviewers 3 [PF2i] and 4 [uwpc]. Again, we reiterate that the improved performance gains are due to AD-GCL's search of non-uniformly random augmentations.

[1] You, Yuning, et al. "Graph Contrastive Learning Automated." arXiv preprint arXiv:2106.07594 (2021).

---

### Decision · Program_Chairs · 2021-09-27

**Decision:**

Accept (Poster)

**Comment:**

The paper combines the ideas of adversarial training (on augmentation) and graph contrastive learning to further boost the unsupervised performance, targeting at representing less redundant information. It is empirically demonstrated with significant performance gains in multiple experiment settings of unsupervised, semi-supervised and transfer learning.

After the discussion, all reviewers agreed that this paper is above the acceptance threshold. The authors are encouraged to improve this work based on the feedback from the reviewers, especially the following ones.

1. In experiments, compare with a recent work -- "Graph Contrastive Learning Automated", which also leverages adversarial training for GCL. In addition, the authors can compare with the following two works:

     - Graph Contrastive Learning with Adaptive Augmentation

      - Graph Contrastive Learning Automated

2. The performance gain of the proposed method is not consistent among datasets, especially when comparing to its direct baseline (GraphCL). This makes the author's claim and the proposed method less convincing. The authors may need more experiments to show the effectiveness of their methods on different tasks.